# ATTRIBUTION-GUIDED EXIT POLICY FOR RELIABLE EARLY-EXIT INFERENCE

## ABSTRACT

Early-exit neural networks enhance efficiency by producing predictions at intermediate layers, but their utility depends critically on the exit policy that decides when to stop inference. Conventional exit policies—often based on confidence thresholds—make effective decisions but do not inherently provide explanations for why a sample exits early. We address this limitation with two contributions. We address this limitation with two contributions. First, we introduce *Progressive Feature Attribution Maps* (PFAM), which capture the network's evolving focus through both exit-wise and cumulative attributions. Second, we propose the *Interpretability-Based Early-Exit Score* (IEES), a composite metric that unifies confidence, attribution progression, and feature relevance. To eliminate attribution overhead at deployment, we further develop a lightweight proxy predictor that estimates IEES from inexpensive, attribution-free features. Experiments on ResNet-18 (CIFAR-10), MSDNet (CIFAR-100), and MobileNetV3 (ImageNet) show that our approach improves accuracy by up to 3% while reducing computation by up to 26% compared to full-depth inference. Analyses demonstrate that IEES-driven exits are semantically coherent and reliable, while PFAM achieves near-perfect monotonicity, convergence alignment, and strong faithfulness across benchmarks. Together, these advances enhance the transparency of anytime prediction and establish a foundation for explainable early-exit inference.

## 1 INTRODUCTION

Deep neural networks (DNNs) achieve state-of-the-art performance across many domains (Alshemali & Kalita, 2020; Fraternali et al., 2023; Dhanjal & Singh, 2024), but their high computational cost limits deployment on resource-constrained platforms. Early-exit DNNs address this by adding intermediate classifiers, allowing inference to stop once predictions are confident (Teerapittayanon et al., 2016), thereby reducing latency and energy use while retaining accuracy. Yet in safety-critical settings such as healthcare or autonomous driving, efficiency alone is not enough—decisions must also be transparent (Rudin, 2019). For example, a chest X-ray prediction may appear confident, but without highlighting disease-relevant regions, clinicians may hesitate to trust it.

Most explainable AI (XAI) methods target single-exit models, producing explanations only at the final layer (Arrieta et al., 2020). Early-exit networks, however, introduce multiple decision points, each requiring justification. Moreover, explanations should capture how model attention evolves across layers—an aspect overlooked by conventional XAI methods (Gilpin et al., 2018).

We address this gap with an explainability-aware early-exit framework built on three components. *Progressive Feature Attribution Maps* (PFAM) visualize how model focus evolves across exits, producing localized and cumulative attributions. The *Interpretability-Based Early-Exit Score* (IEES) quantifies exit reliability by unifying confidence, feature relevance, and attribution stability. A lightweight proxy predictor estimates IEES from attribution-free features at inference, ensuring efficiency. Together, these components enable real-time, explainability-driven exit decisions, which are critical in domains where unreliable early exits can undermine trust.

We evaluate our framework on ResNet-18 (CIFAR-10), MSDNet (CIFAR-100), and MobileNetV3 (ImageNet). Results show consistent improvements in accuracy–efficiency trade-offs over confidence-only baselines, while producing exits aligned with stable, class-relevant explanations.

These findings highlight the potential of attribution-guided exit policies for efficient and trustworthy adaptive inference.

**Contributions.** This work makes the following key contributions:

1. **PFAM:** Progressive attribution maps that capture both exit-wise and cumulative explanations, revealing how model attention evolves across inference stages.

2. **IEES:** A semantic reliability score that integrates confidence, attribution relevance, and stability to assess intermediate exits. For deployment, IEES is approximated by a lightweight proxy predictor that estimates the score from attribution-free features, enabling efficient inference.

3. **Benchmarking:** Extensive comparisons with exit policies and post-hoc attribution methods show that our approach achieves better accuracy–efficiency trade-offs and stronger semantic alignment across exits.

**Paper Organization.** Section 2 reviews related work, Section 3 presents the methodology, Sections 4–5 cover experiments and results, Section 6 compares with recent exit policies, Sections 7–8 provide discussion and limitations, and Section 9 concludes.

## 2 RELATED WORK

**Early-Exit DNNs.** Early-exit architectures improve efficiency by allowing computation to terminate at intermediate layers once an exit criterion is satisfied (Rahmath P et al., 2023; Pacheco et al., 2021). Easy inputs exit early, while harder ones propagate deeper, making these models attractive for latency-constrained settings. BranchyNet (Teerapittayanon et al., 2016) first added intermediate classifiers to AlexNet, LeNet, and ResNet, achieving speedups with minimal accuracy loss. The idea was later extended to CNNs such as MobileNets (Howard et al., 2017), VGG (Kannan et al., 2023), MSDNet (Huang et al., 2017), and EfficientNet (Tan & Le, 2019), as well as to Transformers (Ilhan et al., 2024; Xin et al., 2021), LSTMs (Lattanzi et al., 2023), and GNNs (Rahmath P et al., 2024). Recent extensions span multiple modalities: Demir & Akbas (2024) enhance CNN-based early exits with learnable confidence branches and a unified accuracy–cost objective; in NLP, DeeBERT (Xin et al., 2020) and CEEBERT (Bajpai & Hanawal, 2024) show that intermediate Transformer layers already hold sufficient discriminative power for entropy-based exiting; and parallel decoding frameworks such as FREE (Bae et al., 2023) demonstrate that coordinating shallow and deep predictions can further accelerate early exits in autoregressive models.

**Exit Policies.** Exit policies decide when to stop inference. Classical ones use fixed confidence or entropy thresholds (Pacheco et al., 2021; Yu et al., 2023; Wenjian et al., 2023), while adaptive versions employ reinforcement learning (Wang et al., 2018) or learned controllers (Dai et al., 2020). Recent work explores risk-sensitive exits (Nguyen et al., 2024), unsupervised criteria (Khadem-Sohi et al., 2024), and budget-aware scheduling (Ilhan et al., 2024). ZTW (Singh et al., 2021; 2023) recycles predictions, BEEM (Bajpai & Hanawal, 2025) boosts ensemble margins, and E3 (Lin et al., 2025) uses Integrated Gradients offline to guide allocation, though its runtime exits remain scheduler-driven. Recent advances tighten exit decision quality: CALM (Schuster et al., 2022) enforces sequence-level consistency for reliable early exits in language models, while FREE (Bae et al., 2023) adapts thresholds via a Beta-mixture strategy instead of relying on fixed confidence rules. Uncertainty-focused approaches further refine exit reliability—Meronen et al. (Meronen et al., 2024) correct overconfidence via Laplace-based uncertainty estimates, and CEEBERT (Bajpai & Hanawal, 2024) learns exit thresholds online to improve robustness under domain shift. Nonetheless, most existing policies emphasize accuracy–efficiency trade-offs, with limited attention to the semantic reliability of early predictions. IEES addresses this gap by embedding attribution stability directly into the exit signal.

**Explainable AI.** XAI methods clarify model predictions in critical settings (Gilpin et al., 2018). Saliency maps (Simonyan et al., 2013) and Integrated Gradients (Sundararajan et al., 2017) highlight feature importance, while Grad-CAM (Selvaraju et al., 2017) and Grad-CAM++ (Chattopadhay et al., 2018) yield class-discriminative visualizations. Model-agnostic approaches such as LIME (Ribeiro et al., 2016) and SHAP (Lundberg & Lee, 2017) rely on surrogates or Shapley scores. These methods, designed for single-exit models, provide isolated explanations without capturing how relevance evolves across exits. PFAM and IEES fill this gap by producing exit-wise

and cumulative attributions and embedding stability into the exit signal, enabling stage-wise and semantically reliable decisions.

# 3 METHODOLOGY

We propose an explainability-aware framework for early-exit inference with three components: (i) PFAM, which tracks how model focus evolves across exits, (ii) IEES, which integrates confidence and attribution signals into a reliability score, and (iii) a lightweight *proxy predictor* that estimates IEES from attribution-free features. Unlike prior work that leverages attribution only for training-time adjustments or scheduling (Lin et al., 2025), our framework learns an attribution-grounded reliability score and deploys it via a proxy, ensuring attribution-guided exit decisions at runtime without overhead.

## 3.1 FRAMEWORK OVERVIEW

The pipeline operates in three phases. (a) **Training:** A backbone (e.g., ResNet, MobileNet) is augmented with intermediate classifiers and trained jointly. Complete training details are provided in Section 4. (b) **Calibration:** PFAMs are generated to characterize attribution focus. From these, feature relevance and attribution stability are derived and, together with confidence, define IEES. Weights $(w_1, w_2, w_3)$ and per-exit thresholds $\tau_i^*$ are optimized, and the calibrated IEES scores supervise proxy training on attribution-free features. (c) **Inference:** The trained proxy predicts IEES at each exit. If $\hat{\text{IEES}}_i(x) \geq \tau_i^*$, inference halts; otherwise it proceeds deeper. PFAM maps are optional at test time and used only for visualization or auditing.

## 3.2 PROGRESSIVE FEATURE ATTRIBUTION MAPS (PFAM)

PFAM extends attribution from single predictions to progressive trajectories, showing how model focus evolves across exits. For exit $i$, an attribution map is defined generically as

$$A_i(x) = \text{AttributionMethod}(f_i, x), \tag{1}$$

where $f_i$ denotes the backbone up to exit $i$, mapping the input $x$ to the intermediate feature representation used by that exit. This can be instantiated with any attribution technique (e.g., Grad-CAM, IG, LayerCAM). In our experiments we adopt Grad-CAM for efficiency, yielding

$$A_i(x) = \text{ReLU}\left( \sum_k \gamma_{i,k}\, a_{i,k}(x) \right), \tag{2}$$

where $a_{i,k}(x)$ are activation maps of channel $k$ at exit $i$, and $\gamma_{i,k}$ are their gradient-derived weights. To capture progression, PFAM computes cumulative attribution

$$C_i(x) = \tfrac{1}{i} \sum_{j=1}^{i} A_j(x), \tag{3}$$

and measures stability via the attribution stability score

$$Q_i(x) = \tfrac{1}{|\Omega|} \sum_{p \in \Omega} \big| C_i(x)[p] - C_{i-1}(x)[p] \big|, \tag{4}$$

with $\Omega$ the spatial domain. Small $Q_i(x)$ indicates convergent, reliable focus. PFAM is attribution-agnostic and general, with novelty in its progressive integration and stability analysis.

## 3.3 INTERPRETABILITY-BASED EARLY-EXIT SCORE (IEES)

IEES quantifies the reliability of exit $i$ by combining confidence, attribution strength, and attribution stability:

$$\text{IEES}_i(x) = w_1 S_i(x) + w_2 F_i(x) - w_3 Q_i(x), \tag{5}$$

where

$$S_i(x) = \max_c p_i(c \mid x), \tag{6}$$

denotes the exit confidence. The term $Q_i(x)$ is the attribution stability score defined in Eq. 4. The weights $(w_1, w_2, w_3)$ are tuned on a held-out validation set.

To capture class-relevant evidence, the attribution strength term $F_i(x)$ is defined as:

$$F_i(x) = \frac{1}{HW} \sum_{h=1}^{H} \sum_{w=1}^{W} \left( \sum_{c=1}^{C} w_c A_{h,w,c}(x) \right), \tag{7}$$

where $H$ and $W$ are the spatial dimensions of the feature map, $C$ is the number of channels, $A_{h,w,c}(x)$ is the activation at spatial location $(h, w)$ in channel $c$, and $w_c$ denotes the gradient-derived importance weight for channel $c$. This formulation computes a channel-weighted activation map and then averages it spatially, so activations aligned with class-relevant channels contribute more strongly to the score. A higher $F_i(x)$ indicates stronger, more discriminative attention, re-inforcing the reliability term in IEES by capturing how well an exit's prediction is supported by class-focused evidence.

We term this guiding principle *semantic reliability*: an exit is considered reliable if its prediction is (i) confident, (ii) supported by strong class-relevant activations, and (iii) stable in attribution focus across successive exits. The signs in Eq. 5 follow naturally from this principle—confidence $(+S)$ and relevance $(+F)$ are rewarded, while instability $(-Q)$ is penalized. This principle aligns with prior interpretability research emphasizing stability as a desirable property of explanations (Gilpin et al., 2018). Here, *interpretability* is used in a restricted, operational sense consistent with semantic reliability. IEES integrates these complementary signals into a single score, making it modular and applicable within standard confidence-thresholding or alternative exit controllers. Empirical evidence for these design choices is provided in Appendices G and H, where confidence and attribution strength correlate positively with accuracy, while instability correlates negatively and shows clear separation between exit and continue decisions.

**Calibration.** Calibration proceeds in two stages: (i) optimize weights $(w_1, w_2, w_3)$ via grid search to maximize accuracy–efficiency trade-offs, and (ii) set per-exit thresholds $\tau_i^*$ to balance efficiency and reliability. These thresholds govern inference and supervise proxy training.

### 3.4 PROXY EXIT PREDICTOR

Since computing attributions at inference is costly, we train a lightweight regressor $g_\theta$ at each exit to approximate the IEES score:

$$\hat{\text{IEES}}_i(x) = g_\theta(z_i(x)), \tag{8}$$

where $z_i(x)$ are attribution-free features (e.g., confidence, entropy, activation statistics). Training minimizes the per-exit loss

$$\mathcal{L}_{\text{proxy},i} = \frac{1}{|\mathcal{D}_{val}|} \sum_{x \in \mathcal{D}_{val}} \left( \hat{\text{IEES}}_i(x) - \text{IEES}_i(x) \right)^2. \tag{9}$$

Among Random Forest, XGBoost, and a shallow MLP, Random Forest gives the best accuracy–latency trade-off and is used by default. Proxy fidelity is reported in Appendix B, where the learned predictors achieve $R^2 \geq 0.80$, Spearman $\rho > 0.85$, and MAE $< 0.06$ across datasets, with sub-millisecond latency. For reference, Appendix B also reports that computing attribution maps costs approximately 0.4–0.5 ms per sample, whereas the learned proxy requires less than 0.1 ms per sample and eliminates all attribution overhead during inference.

**Terminology.** We distinguish between the *oracle IEES* (Eq. 5), which uses attribution maps and is employed only offline to supervise proxy training, and the *proxy-based IEES*, $\hat{\text{IEES}}_i(x)$ (Eq. 8), which is deployed at inference using attribution-free features. Throughout the paper, unless otherwise specified, "IEES" refers to this proxy-based exit policy. For clarity, we use $\hat{\text{IEES}}_i$ in equations and simply "IEES" in text and tables.

**Inference.** At runtime, if $\hat{\text{IEES}}_i(x) \geq \tau_i^*$, inference halts and outputs the prediction; otherwise computation continues to deeper exits. In the worst case where no early exits are triggered, runtime approaches backbone latency, with only minor overhead from the lightweight classifiers. Thus efficiency is bounded by the base model.

## 4 EXPERIMENTAL SETUP

**Datasets and Architectures.** We evaluate on CIFAR-10, CIFAR-100, and ImageNet using ResNet-18, MSDNet, and MobileNetV3 backbones, respectively. CIFAR-10/100 contain 50k training and 10k test images ($32 \times 32$); 5k training samples are held out for validation. ImageNet has 1.2M training and 150k validation images ($224 \times 224$), with 25k held out for validation. All backbones are augmented with three side exits (ResNet-18: stages 2/3/4; MSDNet: layers 3/5/9; MobileNetV3: blocks 3/5/7), following BranchyNet and MSDNet conventions. Each exit uses a $3 \times 3$ convolution, adaptive pooling, and a linear classifier.

**Training.** Backbone and exits are trained jointly for 100 epochs using Adam (lr $1 \times 10^{-3}$, cosine decay, batch size 64, standard augmentation—random crop with 4-pixel padding + horizontal flip for CIFAR-10/100; Random Resized Crop 224×224 + horizontal flip for ImageNet). The loss is a weighted sum of exit-specific cross-entropy terms with branch weights decreasing from early to late exits Teerapittayanon et al. (2016).

**Calibration and Proxy.** Calibration (Section 3) tunes weights $(w_1, w_2, w_3)$ and thresholds $\tau_i^*$ via grid search on validation data ($\tau \in [0.5, 0.95]$, step 0.05). The proxy predictor is a Random Forest regressor trained on attribution-free features (confidence, logits, entropy, activation statistics) to approximate the oracle IEES with sub-ms latency. It is trained on the training set with 10% held out for validation, ensuring no test leakage. Extended proxy evaluation is in Appendix B.

**Evaluation Protocols.** We report top-1 accuracy, exit distribution, latency, and AUC (area under the accuracy–FLOPs curve) using the Budgeted Batch Classification (BBC) protocol (Huang et al., 2017). Whereas inference uses per-exit thresholds $\tau_i^*$, BBC applies a single global threshold $\tau^*$ across exits to match FLOPs budgets. Thresholds are tuned on validation data and fixed at test time. Robustness is evaluated under synthetic corruptions (CIFAR-10-C (Hendrycks & Dietterich, 2019)) and stability by perturbing $\tau_i^*$ within $\pm 10\%$.

**Explainability Benchmark.** PFAM is compared with Grad-CAM, Integrated Gradients (IG), SmoothGrad, and Occlusion. Since these baselines target single-exit models, we evaluate them on truncated sub-networks up to each exit $E_i$ (Appendix C), while PFAM operates across all exits. Attribution quality is assessed using three complementary criteria: (i) *semantic progression*—Monotonic Feature Alignment (MFA $\uparrow$) and convergence Intersection-over-Union (IoU $\uparrow$); (ii) *faithfulness*—deletion AUC ($\downarrow$) and insertion AUC ($\uparrow$); and (iii) human preference, measured through a user study. Formal definitions of these metrics are provided in Appendix D, and the user study setup is described in Appendix J.

**Baselines.** We primarily benchmark against confidence-based exits, the de facto standard, in three variants: (i) *Confidence (Optimized)* with thresholds tuned on validation; (ii) *IEES (Optimized)* calibrated the same way for fairness; and (iii) *Confidence (IEES-Matched)*, where thresholds are adjusted to reproduce IEES's exit distribution. We also compare with recent policies: EENet (Ilhan et al., 2024), SelfXit (KhademSohi et al., 2024), Risk-Controlled Exiting (Nguyen et al., 2024), and BEEM (Bajpai & Hanawal, 2025). Dynamic routing methods are complementary, as IEES provides a semantically grounded score that can integrate into such controllers.

**Hardware.** Experiments use an NVIDIA RTX A6000 (48GB) and 3.91 GHz Intel Xeon CPU with PyTorch. Inference latency is measured as average wall-clock time per test sample with batch size 1. Each result is averaged over five independent reruns, reported as mean ± standard deviation.

## 5 RESULTS AND ANALYSIS

We evaluate the proposed framework along four central research questions, covering reliability, efficiency, robustness, and explainability:

Table 1: Exit-wise accuracy and exit distribution across datasets/backbones. Numbers under E1–E4 in the Accuracy block denote accuracy computed only for samples exiting at that exit. *Confidence (IEES-matched)* mimics IEES's exit pattern using confidence thresholds. Metrics are mean $\pm$ std over 5 runs. Overall accuracy aggregates all exited predictions.

| Dataset/ Backbone | Exit Policy | Exit Rate (%) | | | | Accuracy (%) | | | | Overall Acc. | Avg. Time (ms) |
|---|---|---|---|---|---|---|---|---|---|---|---|
| | | E1 | E2 | E3 | E4 | E1 | E2 | E3 | E4 | | |
| CIFAR–10 / ResNet-18 | No–early–exit | – | – | – | 100.00 | – | – | – | 86.91 | 86.91 ± 0.00 | 4.12 ± 0.00 |
| | Conf. (Optimized) | 48.02 | 30.24 | 9.35 | 12.39 | 88.91 | 89.57 | 89.71 | 74.12 | 87.35 ± 0.03 | 3.05 ± 0.02 |
| | **IEES (Optimized)** | **20.18** | **55.92** | **17.26** | **6.64** | **90.01** | **91.12** | **90.04** | **77.01** | **89.77 ± 0.02** | **3.14 ± 0.02** |
| | Conf. (IEES–matched) | 20.18 | 55.92 | 17.26 | 6.64 | 86.13 | 87.42 | 87.06 | 72.89 | 86.13 ± 0.04 | 3.00 ± 0.03 |
| CIFAR–100 / MSDNet | No–early–exit | – | – | – | 100.00 | – | – | – | 76.67 | 76.67 ± 0.00 | 32.04 ± 0.00 |
| | Conf. (Optimized) | 29.30 | 31.94 | 16.10 | 22.66 | 78.63 | 79.06 | 80.33 | 69.44 | 76.95 ± 0.05 | 25.30 ± 0.03 |
| | **IEES (Optimized)** | **20.08** | **23.92** | **28.89** | **27.11** | **81.71** | **83.19** | **81.13** | **66.94** | **77.89 ± 0.03** | **27.59 ± 0.02** |
| | Conf. (IEES–matched) | 20.08 | 23.92 | 28.89 | 27.11 | 77.89 | 79.96 | 79.98 | 65.62 | 75.78 ± 0.07 | 26.94 ± 0.02 |
| ImageNet/ MobileNetV3 | No–early–exit | – | – | – | 100.00 | – | – | – | 74.06 | 74.06 ± 0.00 | 36.63 ± 0.00 |
| | Conf. (Optimized) | 18.56 | 35.95 | 25.42 | 20.07 | 77.01 | 78.02 | 78.10 | 65.04 | 75.04 ± 0.03 | 27.56 ± 0.02 |
| | **IEES (Optimized)** | **16.83** | **25.82** | **26.94** | **30.41** | **81.12** | **80.34** | **81.01** | **68.05** | **76.91 ± 0.01** | **29.49 ± 0.03** |
| | Conf. (IEES–matched) | 16.83 | 25.82 | 26.94 | 30.41 | 76.92 | 75.99 | 77.69 | 66.96 | 74.31 ± 0.04 | 28.53 ± 0.01 |

**RQ1:** Can IEES deliver reliable and explainable exits without compromising performance?
**RQ2:** Does IEES improve accuracy–efficiency trade-offs under compute budgets?
**RQ3:** How robust is IEES to threshold variation and distribution shift?
**RQ4:** How does PFAM compare to standard attribution methods in explaining exit decisions?

### 5.1 RQ1: CAN IEES DELIVER RELIABLE AND EXPLAINABLE EXITS WITHOUT COMPROMISING PERFORMANCE?

We compare IEES with confidence-based exit policies under two settings: (i) *optimized thresholds*, tuned for best accuracy–efficiency trade-offs, and (ii) *matched exit rates*, where confidence thresholds reproduce IEES's sample allocation to isolate the effect of the scoring function. Table 1 shows that IEES consistently improves performance under both settings. On CIFAR-10, IEES routes 76% of inputs by Exit 2 (cumulative across Exits 1–2), achieving 89.77% accuracy at 3.14 ms—outperforming both optimized confidence (87.35%, 3.05 ms) and full inference (86.91%, 4.12 ms). When confidence is constrained to match IEES's exit distribution, accuracy drops to 86.13%, reflecting weaker semantic grounding. In some cases, IEES even surpasses full-depth accuracy, consistent with prior findings that multi-exit supervision provides regularization benefits (Teerapittayanon et al., 2016). On CIFAR-100, IEES achieves 77.89% accuracy (27.59 ms), higher than both optimized confidence (76.95%, 25.30 ms) and matched confidence (75.78%, 26.94 ms). On ImageNet, IEES resolves 70% of inputs by Exit 3, with only 30% requiring full-depth inference. It surpasses full-depth accuracy (76.91% vs. 74.06%) with 20% lower latency; matched confidence drops to 74.31%. Slightly higher latency relative to confidence baselines arises because IEES routes a subset of hard samples to deeper exits—a desirable trade-off, since confident, semantically grounded exits save computation while challenging cases receive additional processing.

Finally, the last exit is naturally biased toward harder inputs, as easier cases are resolved earlier. This phenomenon was explicitly noted in BranchyNet (Teerapittayanon et al., 2016), where easy inputs exited early while more difficult ones were deferred, and further confirmed by MSDNet (Huang et al., 2017), which showed quantitatively that later exits handle disproportionately difficult residual samples. As a result, their accuracy on this subset may appear lower than the backbone's overall accuracy on the full test set. Such behavior is expected and desirable: it indicates that early exits capture simpler cases effectively, thereby improving efficiency without compromising reliability. An ablation study in Appendix I further confirms that confidence, attribution strength, and stability provide complementary, non-redundant contributions to IEES.

**Attribution-Guided Exit Behavior.** Figure 1 illustrates how IEES integrates confidence, attribution strength, and semantic consistency to enable reliable early exits. In Figure 1(a), both signals improve across exits, allowing IEES to stop at Exit 3 with a correct, semantically grounded prediction. Figure 1(b) shows premature overconfidence: attribution refines by Exit 3, but the prediction remains incorrect. A confidence-only policy would exit at Exit 2, while attribution-only reasoning

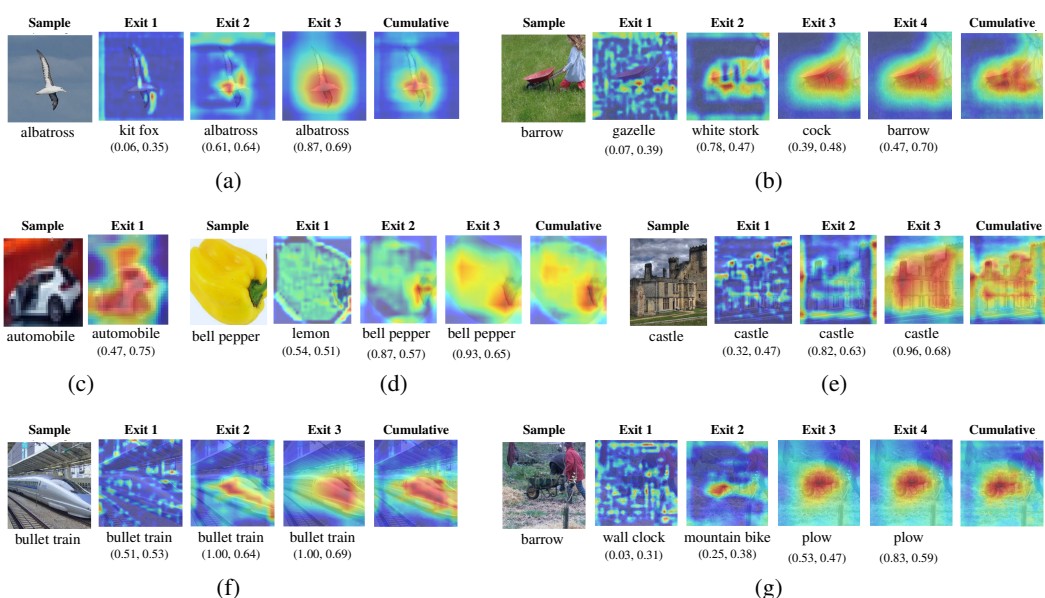

Figure 1: Qualitative examples of attribution maps with ground-truth and predicted labels, along with prediction confidence ($P$) and IEES score ($I$). In each row, the label below the input sample is the ground truth, while labels below exit maps show the predicted class with its corresponding $(P, I)$. (a) Correct exit at Exit 3 with strong confidence and attribution. (b) IEES defers to Exit 4 due to unstable attribution, correcting a premature confidence-only exit. (c) Early exit at Exit 1 based on stable attribution; the cumulative map is omitted as it is identical to Exit 1. (d) IEES avoids premature exit under high but unreliable confidence. (e) Attribution quality refines progressively across exits. (f) Exit is delayed until semantic focus emerges. (g) IEES defers fully for persistently ambiguous inputs.

might stop at Exit 3. By combining both, IEES defers to Exit 4, producing a more trustworthy outcome.

Figure 1(c) demonstrates that IEES can act on strong attribution cues even when confidence is moderate, enabling an accurate early exit at Exit 2. Conversely, Figures 1(d)–1(f) highlight failure modes of confidence-only exits, where high but unreliable confidence leads to premature predictions despite diffuse or misaligned attributions. IEES mitigates this by waiting until attribution maps sharpen around the relevant object or region. Figure 1(e) depicts gradual improvement in attribution quality, resulting in a later but semantically stronger decision. Finally, Figure 1(g) shows a persistently ambiguous sample: although the final output remains incorrect, IEES defers to the deepest exit, reflecting robustness in high-uncertainty scenarios.

Collectively, these case studies highlight the limitation of confidence-only policies: they overlook the semantic quality of explanations. By requiring both confidence and attribution consistency, IEES avoids premature exits and enhances the reliability of early decisions—an essential property for deployment in trust-sensitive domains.

### 5.2 RQ2: DOES IEES IMPROVE ACCURACY–EFFICIENCY TRADE-OFFS UNDER COMPUTE BUDGETS?

We evaluate deployment-time efficiency under the BBC paradigm, where a global FLOPs budget governs early-exit decisions (Section 4). Figure 2 shows FLOPs–accuracy curves for ResNet-18 on CIFAR-10, MSDNet on CIFAR-100, and MobileNetV3 on ImageNet. Across all settings, IEES consistently outperforms confidence-based exits, attaining higher accuracy at equal or lower computational cost. The red marker indicates the optimal budget threshold and the black marker full-depth inference. Quantitatively, IEES yields higher AUC in every case: 0.644 vs. 0.540 (CIFAR-10), 0.539 vs. 0.480 (CIFAR-100), and 0.659 vs. 0.572 (ImageNet). These gains confirm that attribution-aware scoring yields stronger anytime prediction under resource constraints. IEES further reduces latency by exiting earlier on semantically unambiguous inputs—achieving a 24% reduction on CIFAR-10 and 20% on ImageNet—while confidence curves plateau or degrade under stricter budgets. By

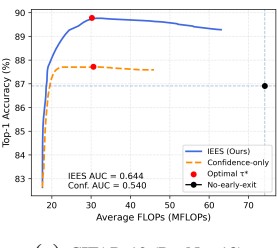 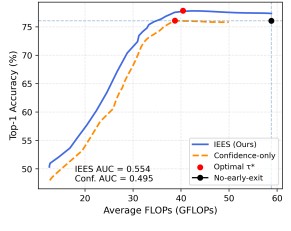 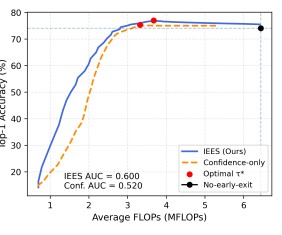

(a) CIFAR-10 (ResNet-18)  (b) CIFAR-100 (MSDNet)  (c) ImageNet (MobileNetV3)

Figure 2: Accuracy–FLOPs trade-off under BBC. IEES consistently outperforms confidence-only policies, achieving higher accuracy at similar or lower computational cost. Red and black dots denote the optimal threshold and full-model inference, respectively. Curves are smoothed for visualization.

leveraging attribution consistency and semantic saliency, IEES delivers more calibrated and trustworthy early exits. Additional visualizations of threshold–accuracy and threshold–exit trade-offs are provided in Appendix F.

### 5.3 RQ3: HOW ROBUST IS IEES TO THRESHOLD VARIATION AND DISTRIBUTION SHIFT?

**Threshold Sensitivity.** To test calibration robustness, we perturbed the decision thresholds at each exit by $\pm 10\%$ around the optimal values $\tau_i^*$. As summarized in Table 4 (Appendix A), IEES maintains stable accuracy and exit behavior across datasets, with deviations typically within $\pm 1.5\%$ accuracy and only minor latency shifts. This indicates that IEES remains reliable under mild miscalibration, a valuable property for deployment where thresholds may not be perfectly tuned.

**Distribution Shift Evaluation.** We next assess robustness under distribution shift using CIFAR-10-C (Hendrycks & Dietterich, 2019), with thresholds calibrated once on clean data and held fixed. Table 3 (Appendix A) reports accuracy and average exit index under common corruptions (severity level 3). IEES naturally adapts by routing more samples to deeper exits as inputs degrade: e.g., the average exit index rises from 2.10 on clean data to 2.91 under motion blur and 3.02 under Gaussian noise. While accuracy declines under these corruptions (73.34% and 68.55%, respectively), the drop is gradual, without retraining or recalibration. This adaptive shift arises from reduced confidence and attribution stability under corruption, which cause IEES to defer decisions to later exits.

Overall, IEES is robust to both threshold perturbations and distributional shift, maintaining stable performance and adapting its exit strategy without additional supervision—supporting its use in dynamic, real-world settings.

### 5.4 RQ4: HOW DOES PFAM COMPARE TO STANDARD ATTRIBUTION METHODS IN EXPLAINING EXIT DECISIONS?

We benchmark PFAM against standard post-hoc attribution methods—IG, SmoothGrad, Grad-CAM, and Occlusion—adapted to operate at individual early exits. Unlike these static techniques, PFAM generates both exit-wise and cumulative heatmaps, capturing how model attention evolves across inference stages. Figure 3 shows that PFAM yields more coherent, class-focused explanations. For example, in Figure 3(a), baselines produce diffuse saliency for a misclassified "cat" sample, while PFAM progressively sharpens focus on the object. Similarly, Figure 3(b) shows PFAM maintaining stable localization across exits. Cumulative maps integrate evidence, offering a trajectory of prediction formation. In Figure 3(c), PFAM remains consistent, while baselines fluctuate, reinforcing model confidence through stable semantic localization. Additional visual comparisons are provided in Appendix K.

Beyond qualitative inspection, PFAM achieves higher scores on semantic progression (Monotonic Feature Alignment and convergence IoU) and faithfulness (deletion and insertion AUCs) than baseline methods. The full quantitative results are reported in Appendix E (Table 6). A small-scale user study (Appendix J) further supports these findings: participants overwhelmingly preferred PFAM's visualizations and judged IEES-triggered exits as semantically reasonable.

Overall, PFAM enhances transparency in anytime prediction by providing progressive, stage-aware explanations that are both quantitatively reliable and human-interpretable. In contrast, standard post-hoc methods offer only isolated snapshots of model behavior.

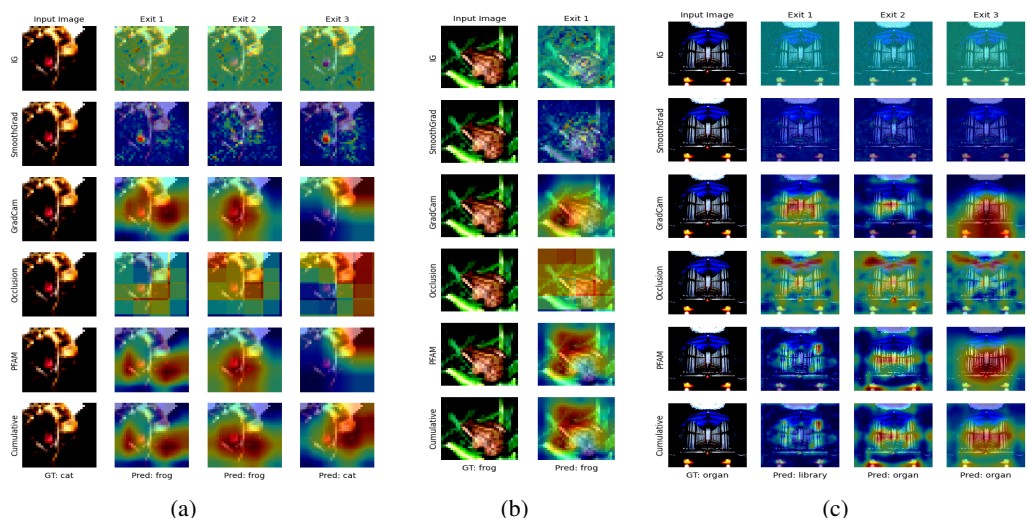

Figure 3: PFAM vs. standard attribution methods: (a) for a misclassified Exit 1 sample, PFAM isolates the error more clearly; (b) for a correct Exit 1 case, it stays focused while others diffuse; (c) across exits, it maintains consistency, reinforcing model confidence.

## 6 COMPARISON WITH RECENT EXIT POLICIES

We benchmark IEES against recent exit policies, including EENet (Ilhan et al., 2024) (budget-aware scheduling), BEEM (Bajpai & Hanawal, 2025) (ensemble margin boosting), ZTW (Singh et al., 2021; 2023) (recycled predictions), E3 (Lin et al., 2025) (deadline-aware allocation), and E2CM (Xu et al., 2022) (class-mean distances). Table 2 summarizes reported FLOPs savings, speedups, and accuracy changes relative to each method's baseline. Across CIFAR-10, CIFAR-100, and ImageNet, IEES complements these directions by jointly improving accuracy (+2–3%) and reducing FLOPs and latency by 15–26% (1.16–1.31× speedup), while uniquely grounding exit decisions in attribution stability and semantic saliency. This integration of efficiency, reliability, and interpretability positions IEES as a practical, general-purpose exit policy for vision tasks.

## 7 DISCUSSION AND INSIGHTS

This work presents a unified framework for efficient and explainable early exits, combining the IEES metric, a lightweight proxy predictor, and PFAM visualizations. IEES integrates confidence, attribution relevance, and progression consistency into a single semantic reliability score, enabling exits that are both accurate and trustworthy. The proxy predictor deploys this score at inference with negligible overhead, while PFAM reveals how evidence evolves across exits, making decisions transparent and quantitatively validated through progression and faithfulness metrics. Experiments across RQ1–RQ4 show that IEES improves decision quality under compute constraints, enables earlier and more accurate exits, and remains robust to threshold variation and distribution shift. Additional analyses, including correlation studies of IEES components (Appendix G) and distributional statistics of exit decisions (Appendix H), further support these design choices. These results position IEES as a practical mechanism for anytime prediction: efficient for edge platforms and interpretable for domains requiring accountability. In safety-critical contexts, additional safeguards—such as fallback mechanisms or abstention strategies—can further reduce residual risk. Where human oversight is required, IEES and PFAM complement rather than replace expert judgment by providing semantically grounded and transparent exit decisions. These results position IEES as a practical mechanism for anytime prediction: efficient for edge platforms and interpretable for domains requiring accountability. We use the term anytime prediction in its standard sense, as in MSDNet and Zero-Time-Waste—denoting the ability to produce valid predictions at multiple computational budgets—even though exits are placed at discrete layers.

Table 2: Comparison of IEES with recent exit policies. $\Delta$ values are relative to baselines in the original papers; "n/a" denotes missing reports. Negative $\Delta$ FLOPs means lower computation; positive (negative) $\Delta$ Acc. indicates accuracy gain (loss).

| Method | Dataset / Backbone | $\Delta$FLOPs (%) | Speedup ($\times$) | $\Delta$Acc. (%) | Exit Signal |
|---|---|---|---|---|---|
| EENet (Ilhan et al., 2024) | CIFAR-10 / ResNet-56 | n/a | 1.34 | −1.0 | Budget-aware confidence scheduling |
| | CIFAR-100 / DenseNet-121 | n/a | 1.36 | −1.0 | |
| | ImageNet / MSDNet-35 | n/a | 1.32 | −0.4 | |
| BEEM (Bajpai & Hanawal, 2025) | GLUE (RTE) / ALBERT | n/a | 1.77 | +0.6 | Boosted ensemble confidence |
| ZTW (Singh et al., 2021; 2023) | CIFAR-10 / MobileNetV2 | −25 | n/a | +0.8 | Recycled confidence predictions |
| | CIFAR-100 / MobileNetV2 | −25 | n/a | +3.3 | |
| E3-Net (Lin et al., 2025) | CIFAR-100 / ResNet-50 | −22 | n/a | +0.1 | Deadline-aware scheduler (IG-guided allocation offline) |
| | ImageNet / MobileNetV2 | −17 | n/a | +0.1 | |
| E2CM (Xu et al., 2022) | CIFAR-10 / ResNet-152 | −25 | n/a | +2.0 | Class-mean distance (nearest centroid) |
| | KMNIST / WideResNet-101 | −20 | n/a | +2.5 | |
| DeeBERT (Xin et al., 2020) | GLUE (MRPC) / RoBERTa | n/a | 2.37 | +0.2 | Entropy threshold |
| | GLUE (QQP) / BERT | n/a | 1.24 | -0.1 | |
| CEEBERT (Bajpai & Hanawal, 2024) | GLUE (MRPC) / BERT-base | n/a | 1.78 | +0.2 | Bandit-based exit selection |
| | GLUE (RTE-QQP) / BERT-base | n/a | 2.15 | +0.1 | |
| **IEES (Ours)** | CIFAR-10 / ResNet-18 | −26 | 1.31 | +2.8 | Confidence + attribution strength and stability |
| | CIFAR-100 / MSDNet | −15 | 1.16 | +1.2 | |
| | ImageNet / MobileNetV3 | −21 | 1.24 | +2.8 | |

## 8 LIMITATIONS AND FUTURE WORK

While IEES and PFAM advance explainability-aware early exits, several limitations remain. PFAM is attribution-agnostic in principle, but our experiments instantiate it with Grad-CAM; systematic validation with alternative methods (e.g., IG, LayerCAM) is left for future work. Our evaluation is restricted to image classification, and extending to other modalities such as text, graphs, or multimodal tasks remains an open direction. IEES currently requires dataset-specific calibration of weights and thresholds; adaptive or meta-learned calibration could improve robustness under distribution shift. Although we analyze robustness under synthetic corruptions, evaluating performance under more realistic distribution shifts and cross-dataset transfer is essential for deployment. Finally, IEES is orthogonal to dynamic routing approaches such as reinforcement learning or bandit controllers, and incorporating semantic reliability into such policies is a promising line of research.

## 9 CONCLUSION

We presented IEES, a semantic reliability score for early exits, implemented with a lightweight proxy predictor and complemented by PFAM attribution maps. The framework unifies efficiency and explainability, enabling reliable exits without attribution overhead while revealing how model focus evolves across layers. Experiments on CIFAR-10, CIFAR-100, and ImageNet show consistent improvements in accuracy–efficiency trade-offs, robustness to threshold variation and distribution shift, and enhanced semantic interpretability. By linking decision quality to attribution stability, IEES advances trustworthy early-exit DNNs and lays a foundation for integrating explainability into adaptive inference. Its low computational cost, semantic grounding, and visual transparency make it well-suited for high-stakes domains such as medical imaging, autonomous driving, and industrial inspection.

## REPRODUCIBILITY STATEMENT

We have taken several steps to ensure reproducibility of our results. All datasets used (CIFAR-10, CIFAR-100, ImageNet-1K, and CIFAR-10-C) are publicly available. Architectures, training settings, calibration procedures, and evaluation protocols are detailed in Section 4. The definition of IEES and its proxy predictor are provided in Section 3 and Appendix B. Extended analyses, including threshold sensitivity, robustness to distribution shift, and attribution baselines, appear in Appendix C–F. Upon publication, we will release code, trained models, and scripts to reproduce all experiments and figures.

## DISCLOSURE OF LLM USAGE

Large language models (e.g., ChatGPT) were used exclusively for language polishing. All technical contributions, experiments, and analyses are entirely the work of the authors.

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

## APPENDIX

This appendix provides extended technical details, derivations, and additional results that support the main paper.

Appendix A reports the supporting tables for RQ3, summarizing threshold sensitivity and robustness under distribution shift. Appendix B provides details of the proxy predictor, including implementation, comparisons of Random Forest, XGBoost, and MLP regressors, and a proxy fidelity analysis. Implementation details for Grad-CAM, Integrated Gradients, SmoothGrad, and Occlusion baselines are given in Appendix C, while the formal definitions of additional explainability metrics (MFA, convergence IoU, and deletion/insertion AUC) are provided in Appendix D. Appendix E extends the attribution stability analysis for RQ4, and Appendix F examines the effect of threshold selection on accuracy–latency trade-offs. Further supporting analyses are included in Appendix G (correlation analysis of IEES components), Appendix H (distributional statistics of exits under different datasets and policies), and Appendix I (ablations of IEES design choices). Finally, Appendix J reports a small-scale user study evaluating PFAM visualizations and IEES-guided exits, while Appendix K presents additional qualitative visual comparisons.

Together, these sections provide the full technical foundation for the results reported in the main paper.

## A  THRESHOLD SENSITIVITY AND ROBUSTNESS (RQ3)

We provide the full tables supporting RQ3, covering threshold perturbation and robustness under distribution shift. These results are presented here for completeness, with detailed interpretation in Section 5.3.

Table 3: IEES performance under distribution shift (CIFAR-10-C; severity level 3). Thresholds are calibrated on clean data and held fixed. See Section 5.3 for interpretation.

| Corruption Type | Accuracy (%) | Avg. Exit Index |
|---|---|---|
| None (Clean) | $89.77 \pm 0.04$ | $2.10 \pm 0.02$ |
| Motion Blur | $73.34 \pm 0.06$ | $2.91 \pm 0.03$ |
| Brightness | $81.22 \pm 0.05$ | $2.63 \pm 0.02$ |
| Gaussian Noise | $68.55 \pm 0.07$ | $3.02 \pm 0.04$ |

## B  PROXY PREDICTOR DETAILS

We evaluated lightweight regressors as candidate proxies to approximate the oracle IEES score at inference. The input feature vector comprised five attribution-free statistics: softmax confidence, logit margin, prediction entropy, and the mean and maximum activations from the exit branch. These features are inexpensive to compute and correlate with semantic confidence and attribution progression. The target output was the calibrated IEES score computed offline at each exit. Proxies were trained on the training set and validated on a held-out 10% split, with fidelity reported on disjoint test data to avoid leakage.

We compared Random Forest (RF), shallow multi-layer perceptrons (MLPs), and XGBoost regressors. Across datasets, RF consistently provided the best fidelity–latency trade-off. For example, an RF with 100 trees (maximum depth 8, minimum leaf size 5) achieved high fidelity with negligible overhead. MLPs exhibited weaker generalization in low-data regimes, and XGBoost required stronger regularization without outperforming RF. We therefore adopt RF as the default proxy.

Table 4: Accuracy (%) and average inference time under threshold perturbation ($\tau_i^* \pm 10\%$). Results are shown for ResNet-18 on CIFAR-10, MSDNet on CIFAR-100, and MobileNetV3 on ImageNet. See Section 5.3 for interpretation.

| Policy (Threshold) | Exit Rate (%) | | | | Per-Exit Accuracy (%) | | | | Overall Acc. | Avg. Time (ms) |
|---|---|---|---|---|---|---|---|---|---|---|
| | E1 | E2 | E3 | E4 | E1 | E2 | E3 | E4 | | |
| **CIFAR-10 (ResNet-18)** | | | | | | | | | | |
| IEES ($\tau_i^* - 10\%$) | 30.72 | 51.44 | 13.15 | 4.69 | 88.02 | 88.96 | 89.44 | 73.55 | 88.01 ± 0.04 | 3.61 ± 0.03 |
| IEES ($\tau_i^*$) | 20.18 | 55.92 | 17.26 | 6.64 | 90.01 | 91.12 | 90.04 | 77.01 | 89.77 ± 0.02 | 3.14 ± 0.02 |
| IEES ($\tau_i^* + 10\%$) | 13.96 | 53.97 | 11.79 | 20.28 | 91.49 | 90.94 | 91.15 | 77.87 | 88.39 ± 0.03 | 3.95 ± 0.01 |
| **CIFAR-100 (MSDNet)** | | | | | | | | | | |
| IEES ($\tau_i^* - 10\%$) | 25.91 | 25.73 | 30.07 | 18.29 | 78.89 | 79.94 | 79.51 | 64.46 | 76.70 ± 0.04 | 26.00 ± 0.05 |
| IEES ($\tau_i^*$) | 20.08 | 23.92 | 28.89 | 27.11 | 81.71 | 83.19 | 81.13 | 66.94 | 77.89 ± 0.03 | 27.59 ± 0.02 |
| IEES ($\tau_i^* + 10\%$) | 18.78 | 22.93 | 26.23 | 32.06 | 81.58 | 82.41 | 81.04 | 66.92 | 77.12 ± 0.04 | 29.81 ± 0.01 |
| **ImageNet (MobileNetV3)** | | | | | | | | | | |
| IEES ($\tau_i^* - 10\%$) | 21.91 | 28.73 | 26.07 | 23.29 | 78.89 | 78.94 | 79.51 | 65.46 | 76.96 ± 0.04 | 27.90 ± 0.05 |
| IEES ($\tau_i^*$) | 16.83 | 25.82 | 26.94 | 30.41 | 81.12 | 80.34 | 81.01 | 68.05 | 76.91 ± 0.01 | 29.49 ± 0.02 |
| IEES ($\tau_i^* + 10\%$) | 14.78 | 21.93 | 22.03 | 41.26 | 82.67 | 79.41 | 81.44 | 67.92 | 75.59 ± 0.04 | 31.81 ± 0.01 |

Table 5 reports fidelity on CIFAR-10, CIFAR-100, and ImageNet, averaged across exits. In all cases, proxies achieve high agreement with oracle IEES, with $R^2 > 0.80$, Spearman $\rho > 0.85$, and MAE below 0.06.

These scores indicate near-perfect agreement with oracle IEES, confirming that attribution-free proxies can faithfully replace attribution computation at inference. When substituted for the oracle, the proxy incurred accuracy differences below 0.3% across datasets, and exit allocations deviated by less than 2% even under distribution shifts (e.g., CIFAR-10-C).

Table 5: Proxy fidelity on held-out test data (averaged across exits). Higher $R^2$ and $\rho$ indicate stronger approximation.

| Dataset / Backbone | $R^2$ | Spearman $\rho$ | MAE | RMSE |
|---|---|---|---|---|
| CIFAR-10 / ResNet-18 | 0.87 | 0.91 | 0.069 | 0.077 |
| CIFAR-100 / MSDNet | 0.84 | 0.88 | 0.081 | 0.092 |
| ImageNet / MobileNetV3 | 0.80 | 0.85 | 0.072 | 0.084 |

In terms of runtime, RF proxy inference adds only $\sim 0.1$ ms per sample, compared to 0.4–0.5 ms for attribution-based IEES computation. This represents more than an order of magnitude reduction in overhead, making proxies highly practical for deployment in early-exit systems where efficiency and reliability must coexist.

## C  BASELINE ATTRIBUTION METHODS

To enable fair comparison with PFAM, we adapted four widely used attribution techniques—Integrated Gradients (IG), SmoothGrad, Grad-CAM, and Occlusion—to operate on early exits. For each exit $E_i$, we constructed a sub-model by truncating the backbone up to that exit's classifier and applied the attribution method independently, targeting the predicted class logit. This produced a separate attribution map for each exit and input.

All baselines were implemented with the Captum library (Kokhlikyan et al., 2020) using default parameters unless otherwise specified. IG used 50 interpolation steps; SmoothGrad sampled 25 noisy variants with Gaussian noise ($\sigma = 0.2$); Occlusion employed $15 \times 15$ sliding windows with stride 8; and Grad-CAM targeted the last convolutional layer before each exit. All heatmaps were resized to $224 \times 224$ and normalized per sample for consistency.

In contrast, PFAM operates natively within the early-exit framework. It reuses intermediate activations already computed during the forward pass and requires only a lightweight backward step at the chosen exits, implemented via registered hooks. Unlike baseline methods, PFAM does not require

constructing truncated sub-models. Instead, it produces both exit-wise and cumulative maps in a single unified procedure, which empirically reduces runtime overhead. Since IEES (or its proxy) governs exit decisions, PFAM visualizations remain optional at test time but can be invoked for auditing or interpretability.

# D  ADDITIONAL EXPLAINABILITY METRICS

We quantify attribution quality across early exits using three families of metrics: (i) *semantic progression*, measured by Monotonic Feature Alignment (MFA, ↑) and convergence IoU to the final attribution (↑); (ii) *faithfulness*, assessed via normalized deletion and insertion AUC (↓/↑).

Our design follows established principles of robustness and consistency in explanations Alvarez Melis & Jaakkola (2018); Yeh et al. (2019) and leverages standard perturbation-based faithfulness protocols Samek et al. (2016); Hooker et al. (2019). Classical pixel-level change metrics, including cumulative $\Delta$-stability Alvarez Melis & Jaakkola (2018), stepwise L1 differences, and Earth Mover's Distance (EMD), often collapse to trivial values for PFAM (e.g., 0 or nan), since PFAM produces progressively normalized maps with stable mass distribution. We therefore do not adopt these metrics.

## D.1  MONOTONIC FEATURE ALIGNMENT (MFA)

*Monotonic Feature Alignment (MFA)* quantifies whether intermediate exit attributions progressively align with the final attribution. Let $A_i$ denote the attribution map at exit $i$, and $A_L$ the final attribution. Similarity is measured via cosine similarity:

$$s_i = 1 - \cos\_\mathrm{dist}(A_i, A_L), \tag{10}$$

where $\cos\_\mathrm{dist}(\cdot)$ is the cosine distance between flattened maps. MFA is then defined as:

$$\mathrm{MFA} = \not\Vdash\big(s_1 \leq s_2 \leq \cdots \leq s_{L-1}\big), \tag{11}$$

which equals 1 if similarity increases monotonically across exits, and 0 otherwise. This reflects the intuition that explanations should refine progressively toward the final attribution map. The idea is conceptually related to monotonic alignment mechanisms in sequence modeling Raffel et al. (2017); Kim et al. (2020), here adapted to progressive attribution refinement.

## D.2  CONVERGENCE IoU TO FINAL

We further measure semantic alignment by computing the overlap between each exit attribution and the final attribution. Let $\mathrm{TopK}(A_i)$ denote the set of top-$k\%$ salient pixels in $A_i$, selected by thresholding at the $(1 - k)$ quantile. The convergence IoU at exit $i$ is:

$$\mathrm{IoU}(A_i, A_L) = \frac{|\mathrm{TopK}(A_i) \cap \mathrm{TopK}(A_L)|}{|\mathrm{TopK}(A_i) \cup \mathrm{TopK}(A_L)|}. \tag{12}$$

We report the average IoU across all non-final exits. Higher values indicate that exit maps converge consistently to the semantic focus of the final attribution. IoU-based measures have been widely used to evaluate saliency map correctness and localization quality Chattopadhay et al. (2018); Bhalla et al. (2023).

## D.3  FAITHFULNESS: DELETION AND INSERTION AUCs

Faithfulness tests whether manipulating features ranked highly by the attribution impacts the prediction in the expected direction Samek et al. (2016); Hooker et al. (2019). For input $x$, class $y$, and attribution map $A$, pixels (or superpixels) are ranked by attribution score and progressively perturbed. At fraction $f \in [0, 1]$, we denote the perturbed input as $x_f$ and the predicted probability as $p_y(x_f)$. The normalized trajectory is:

$$\hat{p}_y(f) = \frac{p_y(x_f) - p_y(x_\mathrm{base})}{p_y(x) - p_y(x_\mathrm{base})}, \tag{13}$$

where $x_{\text{base}}$ is a baseline input (e.g., mean or blurred image). The area under this curve (AUC) is:

$$\text{AUC} = \int_0^1 \hat{p}_y(f)\, df. \tag{14}$$

Lower AUC in deletion (removing salient pixels should reduce confidence) and higher AUC in insertion (adding salient pixels should increase confidence) indicate greater faithfulness of the attribution.

# E    ATTRIBUTION STABILITY ANALYSIS (RQ4)

We now present the full quantitative evaluation corresponding to RQ4, focusing on attribution stability and progression across exits. Our analysis uses the semantic progression metrics defined in Appendix D—Monotonic Feature Alignment (MFA, ↑) and convergence IoU to the final attribution (↑)—as primary indicators of stability. In addition, we report normalized deletion/insertion AUCs to assess attribution faithfulness. Together, these metrics quantify whether attributions evolve progressively, align with the final attribution map, and highlight features that are causally important for model predictions.

Table 6: Attribution stability and faithfulness metrics across datasets. Higher MFA and IoU, lower deletion AUC, and higher insertion AUC indicate more reliable attributions. Values are mean ± std over 100 samples.

| Dataset / Backbone | Metric | Grad-CAM | IG | SmoothGrad | Occlusion | **PFAM** |
|---|---|---|---|---|---|---|
| CIFAR-10 / ResNet-18 | MFA (↑) | 1.00 ±0.00 | 0.75 ±0.44 | 0.74 ±0.44 | 0.91 ±0.29 | **0.96 ±0.02** |
| | IoU (↑) | 0.20 ±0.00 | 0.65 ±0.07 | 0.36 ±0.06 | 0.71 ±0.15 | **0.93 ±0.02** |
| | Deletion AUC (↓) | 0.35 ±0.11 | 0.19 ±0.19 | 0.31 ±0.15 | 0.25 ±0.14 | **0.28 ±0.14** |
| | Insertion AUC (↑) | 0.55 ±0.13 | 0.58 ±0.26 | 0.56 ±0.17 | 0.62 ±0.16 | **0.63 ±0.18** |
| CIFAR-100 / MSDNet | MFA (↑) | 0.53 ±0.50 | 0.42 ±0.50 | 0.48 ±0.50 | 0.62 ±0.49 | **0.95 ±0.02** |
| | IoU (↑) | 0.30 ±0.13 | 0.50 ±0.12 | 0.33 ±0.11 | 0.58 ±0.23 | **0.92 ±0.03** |
| | Deletion AUC (↓) | 0.16 ±0.15 | 0.15 ±0.17 | 0.17 ±0.14 | 0.17 ±0.14 | **0.14 ±0.15** |
| | Insertion AUC (↑) | 0.46 ±0.20 | 0.27 ±0.23 | 0.43 ±0.19 | 0.45 ±0.17 | **0.50 ±0.21** |
| ImageNet / MobileNetV3 | MFA (↑) | 0.63 ±0.49 | 0.51 ±0.50 | 0.88 ±0.33 | 0.37 ±0.49 | **0.97 ±0.01** |
| | IoU (↑) | 0.21 ±0.09 | 0.31 ±0.06 | 0.32 ±0.05 | 0.27 ±0.09 | **0.91 ±0.02** |
| | Deletion AUC (↓) | 0.17 ±0.09 | 0.11 ±0.06 | 0.14 ±0.09 | 0.12 ±0.08 | **0.14 ±0.12** |
| | Insertion AUC (↑) | 0.29 ±0.11 | 0.16 ±0.10 | 0.29 ±0.13 | 0.31 ±0.13 | **0.40 ±0.15** |

**Results.**    Across CIFAR-10, CIFAR-100, and ImageNet, PFAM achieves consistently higher MFA and convergence IoU than baseline attribution methods. For example, on CIFAR-100 with MSDNet, PFAM reaches MFA = 0.95 and IoU = 0.92, whereas baselines range between 0.3–0.6. On ImageNet, PFAM similarly attains MFA = 0.97 and IoU = 0.91, surpassing all alternatives. These representative results show that PFAM produces attribution maps that evolve smoothly and converge semantically toward the final attribution, unlike Grad-CAM, IG, SmoothGrad, and Occlusion, which often exhibit fluctuating or inconsistent progression.

Faithfulness metrics further confirm that PFAM achieves consistently competitive or superior scores, with lower deletion AUC and higher insertion AUC, indicating that the identified features are causally important for model predictions. For instance, on CIFAR-10 with ResNet-18, PFAM achieves an insertion AUC of 0.63 compared to 0.55 for Grad-CAM and 0.56 for SmoothGrad, while maintaining a deletion AUC comparable to the best baselines.

Overall, these findings quantitatively validate PFAM's qualitative property of progressive refinement across exits. Attribution stability analysis demonstrates that PFAM provides progressive, semantically consistent, and faithful explanations across exits—properties that are highly desirable for reliable interpretability-aware exit policies.

# F    EFFECT OF THRESHOLD SELECTION ON ACCURACY AND EXIT DISTRIBUTION

To complement the quantitative sensitivity results in Section 5.3, we visualize how varying the decision threshold affects IEES accuracy and exit allocation. Figure 4(a) shows the threshold–accuracy

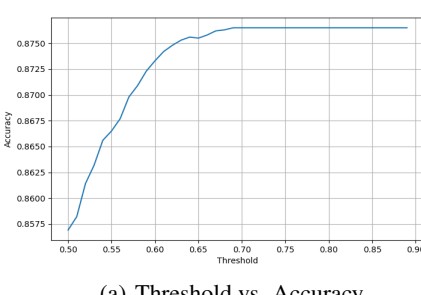
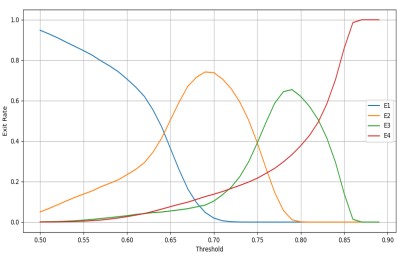

(a) Threshold vs. Accuracy            (b) Threshold vs. Exit Rate

Figure 4: Effect of threshold variation on IEES performance for ResNet-18 on CIFAR-10. (a) Accuracy increases with stricter thresholds until saturation near the optimal point. (b) Exit distribution shifts progressively deeper as thresholds rise, reflecting increased decision confidence.

curve on CIFAR-10 (ResNet-18): increasing thresholds delays early exits, improving accuracy up to an optimal point ($\tau^* = 0.72$), beyond which further gains plateau. Figure 4(b) shows the corresponding change in exit behavior: lower thresholds trigger earlier exits, while higher thresholds defer decisions to deeper branches.

These monotonic trends confirm that IEES provides controllable trade-offs between efficiency and reliability. In practice, calibration enables balancing predictive accuracy, interpretability, and computational cost by selecting thresholds that align with deployment requirements.

## G  CORRELATION ANALYSIS OF IEES COMPONENTS

IEES integrates three complementary signals—confidence ($S$), attribution strength ($F$), and stability ($Q$)—to ensure accurate, feature-grounded, and reliable early exits. Correlation matrices (Figure 5) across CIFAR-10, CIFAR-100, and ImageNet validate the expected relationships of these components with prediction accuracy and with each other.

**Key Observations:**

- **Confidence ($S$):** Confidence exhibits a strong positive correlation with accuracy across all datasets (e.g., 0.62 on CIFAR-10, 0.58 on CIFAR-100, and 0.69 on ImageNet), confirming its role as the dominant signal in guiding reliable exits. As expected, $S$ is positively associated with $F$ (0.42–0.52) and negatively with $Q$ ($-0.30$ to $-0.38$), showing that confident predictions tend to coincide with strong features and stable attributions.

- **Attribution Strength ($F$):** Attribution strength also aligns positively with accuracy (0.38 on CIFAR-10, 0.44 on CIFAR-100, 0.51 on ImageNet). Although its effect is weaker than $S$, it complements confidence by ensuring that high-confidence predictions are grounded in class-relevant activations. Its moderate negative correlation with $Q$ ($-0.22$ to $-0.27$) further suggests that stronger activations tend to be more stable, helping to avoid premature exits.

- **Stability ($Q$):** Stability correlates negatively with accuracy ($-0.28$, $-0.33$, and $-0.37$), consistent with its design as a safeguard against unstable exits. By penalizing predictions with fluctuating attribution patterns, $Q$ ensures that IEES favors semantically consistent exits even under high confidence.

**Dataset-Specific Trends:** The strength of correlations varies across datasets. On CIFAR-10, values are moderate, reflecting the lower complexity of the dataset. CIFAR-100 shows clearer separation between signals, while ImageNet displays the strongest alignment with the IEES design, with both $S$ and $F$ strongly predictive and $Q$ acting as a reliable negative indicator. This progression highlights the importance of dataset-adaptive thresholding to balance accuracy, efficiency, and semantic reliability.

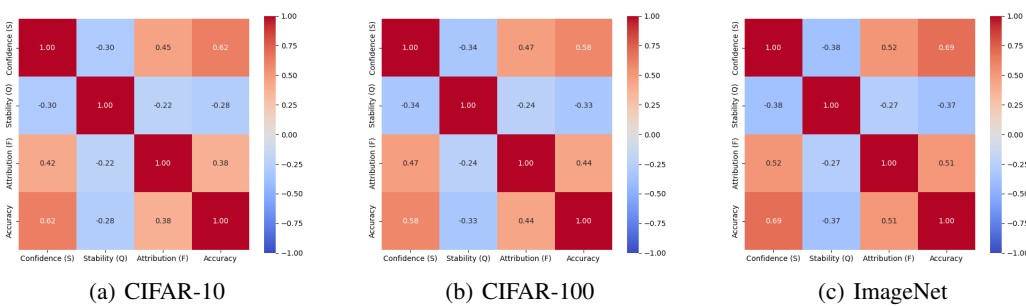

(a) CIFAR-10        (b) CIFAR-100        (c) ImageNet

Figure 5: Correlation matrices of IEES components across CIFAR-10, CIFAR-100, and ImageNet. Confidence and attribution strength are positively correlated with accuracy, while stability shows the expected negative correlation, confirming their complementary roles in guiding early-exit decisions.

**Conclusion:** The three IEES components play distinct yet complementary roles: confidence drives correctness, attribution strength reinforces feature grounding, and stability guards against unreliable early exits. Their integration provides a principled basis for dynamic exit decisions across datasets of varying complexity.

# H    DISTRIBUTION ANALYSIS OF IEES COMPONENTS

We analyzed the distributions of confidence, attribution strength, and attribution stability across "Exit" and "Continue" decisions (Figures 6–8). This analysis highlights how each component contributes to guiding early exits, complementing one another to ensure reliability, interpretability, and stability.

**Key Observations:**

- **Confidence as an Initial Trigger:** Confidence distributions show clear separation at high values, confirming that highly confident samples are more likely to exit early. However, substantial overlap in the mid-range (0.4–0.7) creates an "ambiguity zone," where confidence alone may cause premature or unstable exits. This underscores the need for auxiliary signals.

- **Attribution Strength as a Refinement Signal:** Attribution strength (see Figures 6(b), 7(b), 8(b)) refines reliability by ensuring that early exits occur only when features are strongly activated and semantically discriminative. Its effect is dataset-dependent: sharper separation in CIFAR-10, broader distributions in CIFAR-100 (reflecting higher task complexity), and consistently higher thresholds in ImageNet.

- **Attribution Stability as a Reliability Filter:** Attribution stability (see Figures 6(c), 7(c), 8(c)) serves as a safeguard against unstable exits. Samples with low values (stable attributions) align with Exit decisions, whereas high values (unstable attributions) align with Continue decisions, reflecting uncertainty. This behavior reinforces stability's role as a necessary constraint on early exits.

**Interactions and IEES Justification:** The joint behavior of these components explains the effectiveness of IEES. Confidence drives exits but becomes unreliable in ambiguous regions. Attribution strength complements it by requiring feature saliency, preventing exits on spurious high-confidence activations. Attribution stability enforces semantic consistency, delaying exits when explanations fluctuate. Together, they yield a composite score that is more robust and interpretable than confidence-only baselines.

**Dataset-Specific Thresholding:** Exit behaviors vary across datasets. In CIFAR-10, sharp separation in confidence and attribution strength enables earlier exits. In CIFAR-100, broader distributions reflect increased difficulty, requiring stronger activations before exiting. On ImageNet, thresholds shift upward across all three components, consistent with higher semantic complexity and deeper

architectures. These trends demonstrate the need for dataset-adaptive thresholds rather than fixed exit criteria.

**Conclusion:** Distribution analysis empirically confirms that confidence alone is insufficient for stable early exits. Attribution strength grounds exits in semantically relevant activations, while attribution stability prevents unreliable decisions under fluctuating explanations. Dataset-dependent distribution patterns further show that IEES generalizes effectively across architectures and datasets, unlike static confidence-based methods.

# I    ABLATION OF IEES COMPONENTS

IEES integrates three complementary signals: (i) Confidence ($S$), capturing exit-wise certainty; (ii) Attribution strength ($F$), ensuring decisions are supported by strong class-relevant activations; and (iii) Attribution stability ($Q$), penalizing unstable predictions across exits. To assess their individual contributions, we ablated components on CIFAR-10 (ResNet-18). Thresholds for each variant were tuned on a held-out validation set, and results are averaged over 5 runs (mean $\pm$ std).

Results show that confidence alone yields the lowest performance. Adding stability ($S+Q$) improves accuracy by filtering out unstable exits, though at a modest latency cost. Adding attribution strength ($S+F$) also improves accuracy and slightly reduces latency, since strongly activated features enable earlier safe exits. The full combination ($S + F + Q$) achieves the best overall performance, demon-

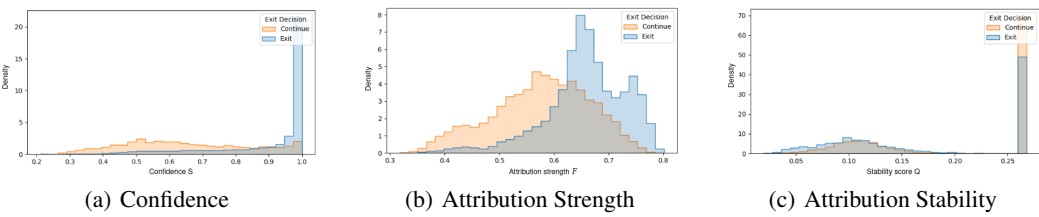

(a) Confidence                (b) Attribution Strength                (c) Attribution Stability

Figure 6: Distribution analysis of IEES components on CIFAR-10 using ResNet-18 (confidence, attribution strength, and attribution stability) across exit decisions (Exit vs. Continue).

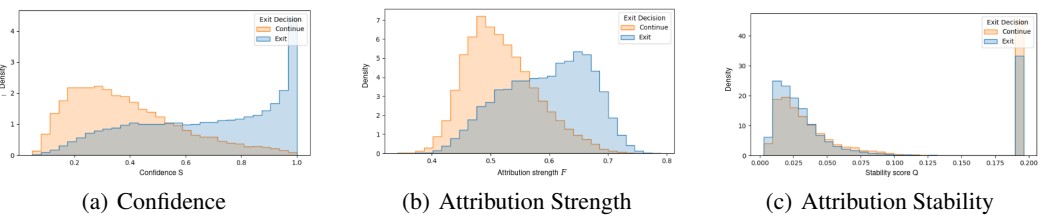

(a) Confidence                (b) Attribution Strength                (c) Attribution Stability

Figure 7: Distribution analysis of IEES components on CIFAR-100 using MSDNet (confidence, attribution strength, and attribution stability) across exit decisions (Exit vs. Continue).

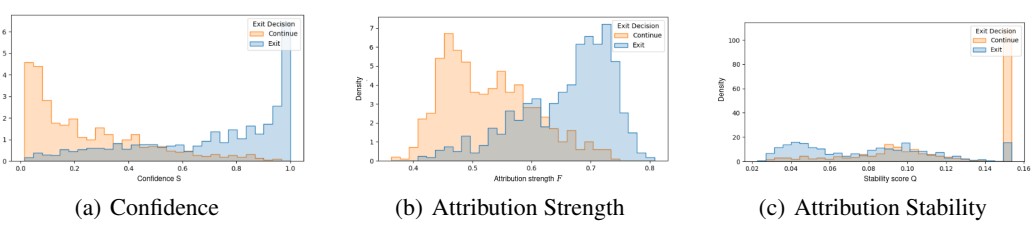

(a) Confidence                (b) Attribution Strength                (c) Attribution Stability

Figure 8: Distribution analysis of IEES components on ImageNet using MobileNetV3 (confidence, attribution strength, and attribution stability) across exit decisions (Exit vs. Continue).

Table 7: Ablation of IEES components on CIFAR-10 (ResNet-18). Accuracy and latency are averaged over 5 runs. Latency differences reflect earlier or deferred exit allocations depending on the available signals.

| Exit Signal | Accuracy (%) | Latency (ms) |
|---|---|---|
| Confidence only ($S$) | $87.35 \pm 0.03$ | $3.05 \pm 0.02$ |
| Confidence + Stability ($S + Q$) | $88.90 \pm 0.04$ | $3.20 \pm 0.03$ |
| Confidence + Attribution ($S + F$) | $88.10 \pm 0.03$ | $2.98 \pm 0.02$ |
| Full IEES ($S + F + Q$) | $\mathbf{89.77 \pm 0.02}$ | $\mathbf{3.14 \pm 0.02}$ |

strating that the three signals are complementary and jointly enhance both accuracy and efficiency compared to confidence-only policies.

## I.1 ABLATION: LOGIT-BASED STABILITY VS. ATTRIBUTION-BASED STABILITY

Stability is a central component of the IEES formulation. To isolate the contribution of attribution-level stability, we evaluate a natural alternative that replaces attribution-based stability with consistency measured directly in logit space. This logit-stability variant retains the overall score structure (confidence term and attribution-strength term), modifying only the stability component:

$$Q_i^{\text{logit}}(x) = \frac{\mathbf{z}_i(x)^\top \mathbf{z}_{i-1}(x)}{\|\mathbf{z}_i(x)\|_2 \, \|\mathbf{z}_{i-1}(x)\|_2}.$$

Both variants are calibrated using the same validation-based grid search to select exit thresholds. Results across CIFAR-10, CIFAR-100, and ImageNet are summarized in Table 8. Logit-stability provides moderate gains over confidence-only routing, indicating that prediction-space smoothness is a useful signal. However, it consistently underperforms IEES, which incorporates stability in the model's explanatory patterns rather than only in its logits. These results show that explanation-level consistency offers a stronger and more reliable criterion for early-exit decisions.

Table 8: Ablation comparing IEES with attribution-based stability to a logit-stability variant. All components except the stability term remain identical.

| Dataset / Backbone | Method | E1 | E2 | E3 | E4 | Overall Acc. | Time (ms) |
|---|---|---|---|---|---|---|---|
| CIFAR-10 / ResNet-18 | Logit-Stability | 89.30 | 89.85 | 89.60 | 75.00 | 88.05 | 3.07 |
| | **IEES (Attrib-Stability)** | 90.01 | 91.12 | 90.04 | 77.01 | **89.77** | 3.14 |
| CIFAR-100 / MSDNet | Logit-Stability | 80.00 | 80.80 | 80.70 | 64.10 | 77.45 | 25.55 |
| | **IEES (Attrib-Stability)** | 81.71 | 83.19 | 81.13 | 66.94 | **77.89** | 27.59 |
| ImageNet / MobileNetV3 | Logit-Stability | 77.10 | 77.50 | 77.60 | 65.70 | 75.00 | 27.65 |
| | **IEES (Attrib-Stability)** | 81.12 | 80.34 | 81.01 | 68.05 | **76.91** | 29.49 |

## J USER STUDY ON ATTRIBUTION QUALITY

To complement quantitative metrics, we conducted a small-scale user study with 12 participants who had prior experience with visual recognition tasks (e.g., computer vision coursework or research). The study aimed to assess whether PFAM heatmaps provided clearer and more semantically aligned explanations than standard attribution methods, and whether IEES-triggered early exits appeared reasonable.

Participants were presented with samples from CIFAR-10, CIFAR-100, and ImageNet that were randomly selected in a class-stratified manner. Each sample included the model prediction, PFAM visualizations (exit-wise and cumulative), and matched baselines (Grad-CAM, IG, SmoothGrad, Occlusion). Participants were asked to choose which heatmap best localized class-relevant regions and to evaluate whether the corresponding exit decision appeared reasonable.

**Results.** PFAM was preferred in between 70% and 80% of comparisons across datasets, and IEES exit decisions were judged semantically reasonable in approximately 90% of cases. These find-

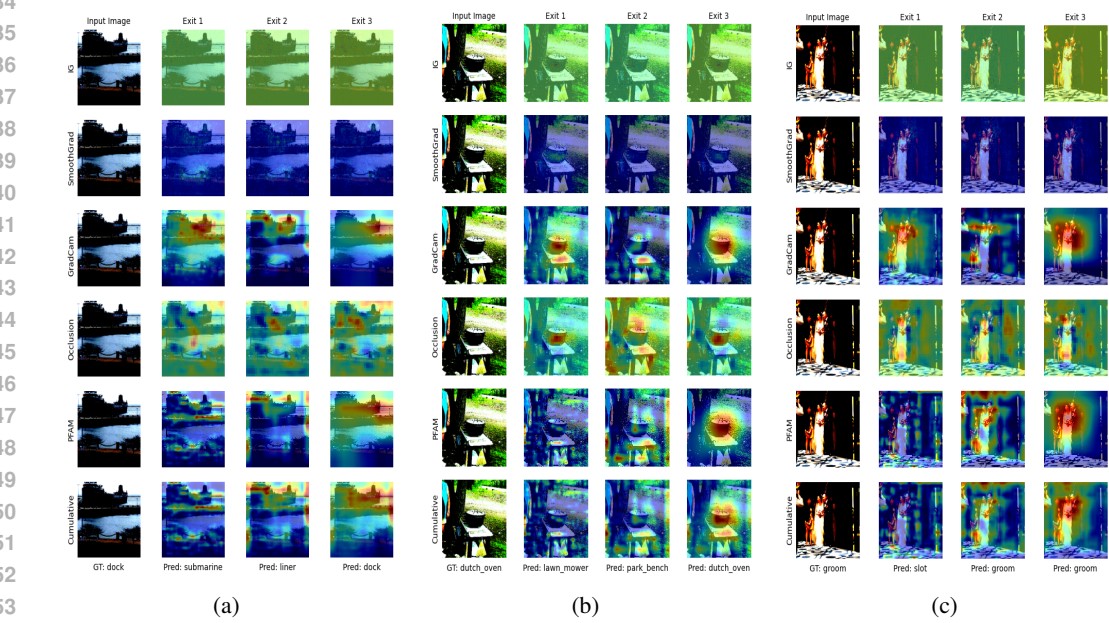

Figure 9: Extended comparison of PFAM with standard attribution methods (IG, SmoothGrad, Grad-CAM, Occlusion). PFAM consistently captures class-relevant regions across exits and provides more coherent visual explanations than baselines.

ings provide initial human-centered validation that PFAM yields clearer explanations and that IEES benefits from attribution-aware exit signals.

All participants provided informed consent, no personal data were collected, and only non-sensitive tasks were involved. The study was conducted in accordance with standard ethical guidelines.

## K  EXTENDED VISUAL COMPARISON: PFAM VS. TRADITIONAL ATTRIBUTION METHODS

Figures 9–11 provide extended visual comparisons of PFAM against standard attribution methods (IG, SmoothGrad, Grad-CAM, Occlusion) across different exits. These examples expand upon Figure 3 in Section 5.4, illustrating attribution quality, prediction correctness, and explanation consistency over depth.

**Misclassification and Attribution Stability.** Examples in Figure 9 show PFAM's ability to produce focused, exit-wise consistent attributions even under misclassification. In Figure 9(a), the model misclassifies at Exit 1; while IG and Grad-CAM produce diffuse or misplaced attention, PFAM highlights the incorrect focus clearly, making the failure interpretable. In Figure 9(b), the model classifies correctly at Exit 1, but PFAM maintains stable attribution across exits, whereas Smooth-Grad yields fragmented heatmaps. Figure 9(c) further demonstrates how PFAM preserves attention alignment throughout all exits, reinforcing prediction confidence.

**Spatial Precision and Progressive Refinement.** As shown in Figure 10, PFAM provides equal or better spatial precision compared to Grad-CAM, particularly at deeper exits where Grad-CAM often loses focus. In Figure 10(a), PFAM captures fine-grained class-specific attention aligned with salient features. In Figure 10(b), it reflects progressive refinement across exits, whereas IG often shows inconsistent drift. Figure 10(c) illustrates an Exit 2 case where PFAM produces clearer, more coherent maps than Occlusion and other baselines.

**Robustness in Challenging Inputs.** Figure 11 highlights PFAM's robustness under challenging inputs. In Figure 11(a), PFAM retains focus on semantically relevant regions while other methods introduce attribution noise. In fine-grained or ambiguous cases such as Figure 11(b) and Figure 11(c),

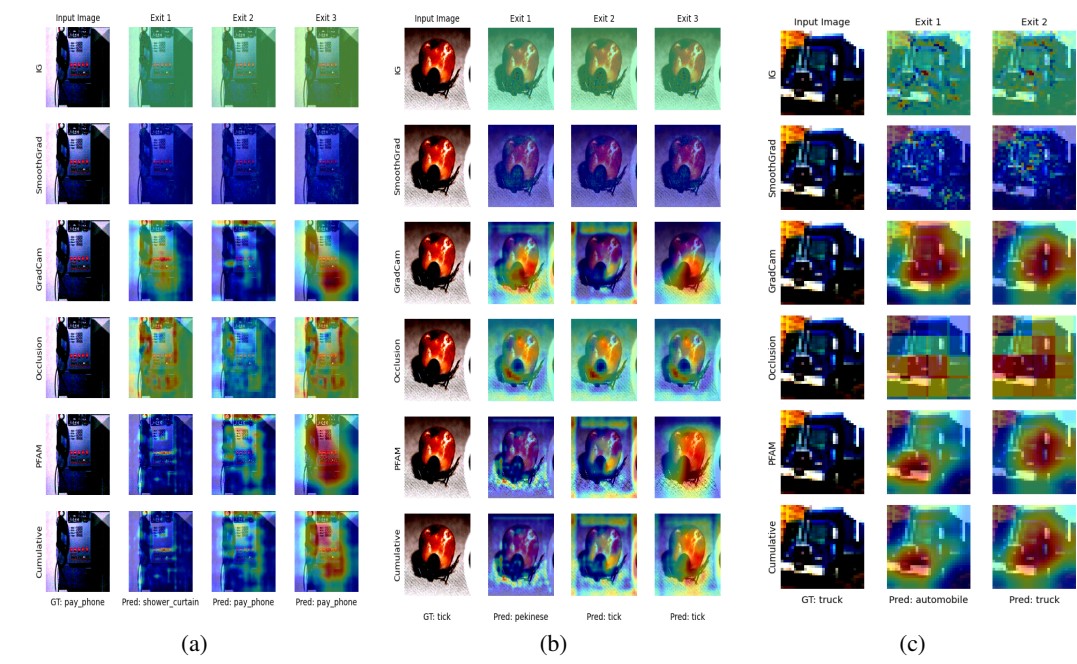

Figure 10: Comparison of attribution quality across exits. PFAM yields stable and progressively refined attributions aligned with model depth, outperforming baseline methods.

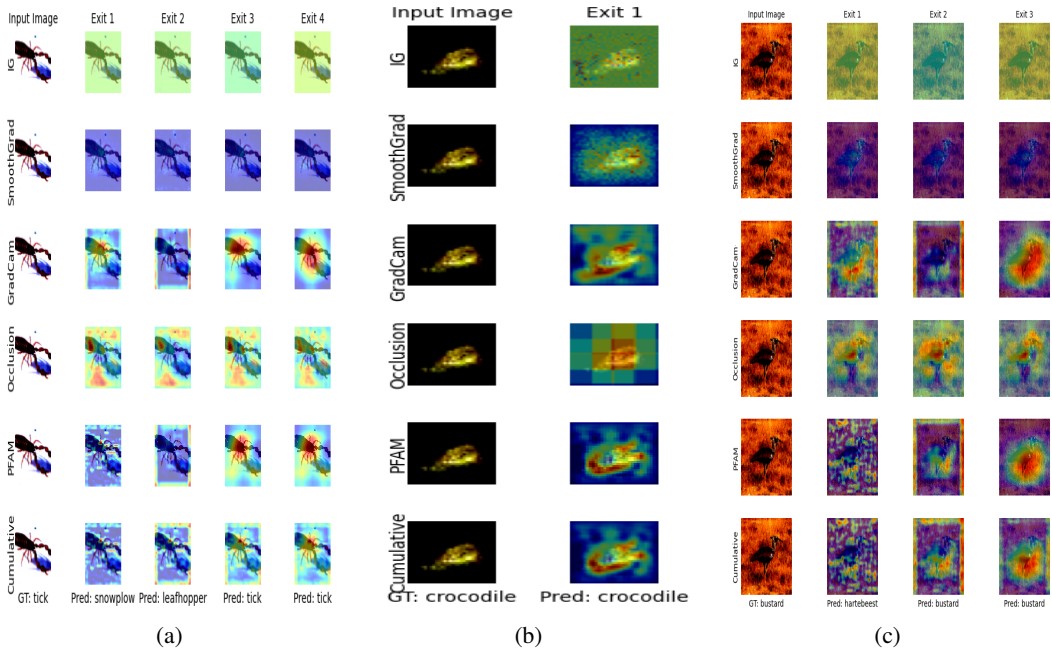

Figure 11: Examples under challenging or ambiguous predictions. PFAM maintains semantic focus and attribution consistency where traditional methods often degrade.

PFAM exhibits sharper and more confident attributions, maintaining semantic consistency as depth increases.

**Cumulative Attribution Mapping.** PFAM uniquely supports cumulative attribution maps that consolidate attention across all exits. These maps visualize how model focus converges toward correct predictions, reflecting the evolving semantic understanding—functionality not supported by traditional post-hoc methods.

**Summary.** Together, these examples show that PFAM matches existing attribution tools in clarity and frequently surpasses them in stability, semantic refinement, and progressive explainability across early-exit networks. Its cumulative maps further capture how focus evolves with depth, offering transparent explanations for both correct and erroneous decisions.

## L  ADDITIONAL CALIBRATION DETAILS

IEES calibration is performed entirely post-training using a held-out validation split and requires only forward passes. The process proceeds in two stages. First, the component weights $(w_1, w_2, w_3)$ are selected through a coarse grid search to establish a stable combined score that balances confidence, attribution relevance, and attribution stability. Once these weights are fixed, we tune the exit thresholds $\tau_i$ for each exit by sweeping values in $[0.5, 0.95]$ using a step of $0.05$ and selecting the threshold that maximizes validation accuracy. Separating the two steps avoids instability that can arise when thresholds are optimized over uncalibrated scores. All calibration is performed strictly on the validation split, and no calibration data is used during testing, ensuring a clean separation between tuning and evaluation.

