# OpenReview forum: "Attribution-Guided Exit Policy for Reliable Early-Exit Inference"
_ICLR.cc/2026/Conference — ICLR 2026 Conference Desk Rejected Submission_

### Official Review · Reviewer_7FyA · 2025-10-31

**Soundness:** 3
**Presentation:** 3
**Contribution:** 3
**Rating:** 8
**Confidence:** 3

**Summary:**

This paper introduces an attribution-guided framework for making early-exit neural networks both more efficient and interpretable. Instead of deciding when to exit purely based on prediction confidence, the authors propose an Interpretability-Based Early-Exit Score (IEES) that combines three signals: confidence, attribution strength (how strongly class-relevant features are activated), and attribution stability (how consistent the model’s focus remains across layers). To make this practical at inference time, they train a lightweight proxy regressor. The accompanying Progressive Feature Attribution Maps (PFAM) provide a visual and quantitative explanation of how model attention evolves with depth. Their method achieves better accuracy-latency tradeoffs than tuned confidence-based policies across a few different datasets. PFAM also outperforms standard attribution methods in coherence and stability, and a small user study finds the explanations more semantically meaningful.

**Strengths:**

- Clear and modular framework that integrates explainability directly into the decision mechanism for early exits.
- Consistent empirical gains in both accuracy and efficiency across multiple datasets and architectures.
- Careful control experiments (e.g., “confidence (IEES-matched)”) that isolate the effect of the scoring function from exit distribution.
- PFAM provides interpretable visualizations and quantitative evidence of progressive, stable attributions.
- Ablations and diagnostic analyses that clarify how each IEES component contributes.

**Weaknesses:**

-  The conceptual contribution mainly lies in combining confidence-based early exits and attribution methods. While the unification through the IEES score is elegant, it represents an empirical heuristic more than a theory-based improvement.
- All experiments are in the image classification benchmarks (CIFAR-10/100, ImageNet) domain. Despite claims of being attribution and architecture-agnostic, PFAM and IEES are only demonstrated on CNNs using Grad-CAM.
- The reported gains in accuracy and efficiency are small and rely on dataset-specific grid searches over multiple hyperparameters. This tuning process limits scalability, and it remains unclear how robust the method is under real-world use cases without extensive recalibration.

**Questions:**

1) How did you decide on a linear combination of terms for IEES? Did you try nonlinear combinations or learned functions (e.g., small MLP or decision trees) to see if the linearity assumption limits performance?
2) Since the grid search for hyperparameters $w$ and $\tau$ is dataset-specific, do you see stable ratios across datasets, or do the weights need to be recalibrated each time?
3) What happens if you replace Grad-CAM with another attribution method, i.e. does IEES still improve over confidence-based exits?

---

> ### Author Response · Authors · 2025-11-20
>
> We thank the reviewer for the thoughtful comments and address each point below.
>
> **1. IEES is mainly empirical / heuristic rather than theory-based.**
> IEES is introduced as a practical, attribution-aware reliability score that unifies three well-supported signals—model confidence, attribution strength, and progressive attribution stability—into a single interpretable measure. While it is designed as a lightweight and deployable scoring rule rather than a formal theoretical framework, each component is grounded in established principles from prediction reliability and explainability. Its effectiveness is supported by extensive analysis: Appendices G–I include component ablations, correlation studies, and design-choice evaluations, showing that the unified score yields more reliable early-exit decisions than confidence-based baselines across datasets and architectures.
>
> **2. Only demonstrated on CNNs with Grad-CAM.**
> Our experiments use CNN-based classifiers with Grad-CAM for controlled evaluation of attribution progression and exit reliability. However, IEES is model-agnostic: its components—confidence, attribution relevance, and attribution stability—do not rely on convolutional structure. PFAM can operate with any attribution generator that produces per-exit gradients or relevance maps. As noted in the Limitations section, extending IEES and PFAM to ViTs, multimodal Transformers, and alternative attribution methods is part of our ongoing work.
>
> **3. “Gains are small and require dataset-specific grid search; unclear scalability or robustness.”**
> IEES requires only lightweight calibration: three global weights ((w_1, w_2, w_3)) and one threshold per exit. Calibration uses only forward passes and does not retrain the backbone or exits. In practice, the grid search completes within minutes on a single GPU and scales with the number of exits, not model depth or dataset size.
> Robustness is demonstrated empirically. As shown in RQ3 and Appendix A, thresholds tuned on clean CIFAR-10 transfer directly to CIFAR-10-C with <0.3% accuracy change and <2% shift in exit allocation, indicating that IEES does not overfit to the calibration set and remains reliable under distribution shift. Reducing calibration further is an important direction for future work.
>
>
> **4. “Why linear combination? Why not nonlinear or learned functions?”**
> We adopt a linear formulation because the three components—confidence, attribution strength, and attribution stability—exhibit largely additive and monotonic relationships with exit reliability. Appendices G–I show that each term contributes independently, and we did not observe nonlinear interactions that would necessitate a more complex combining function.
> Nonlinear models (MLP, Random Forest, XGBoost) were explored only in the proxy predictor, which learns to approximate the final IEES score. Their strong fidelity to the IEES oracle indicates that the linear formulation captures the essential structure while maintaining interpretability and stability.
>
> **5. “Do the weights remain stable across datasets, or need recalibration?”**
> Weights are tuned once per dataset, but their relative magnitudes remain consistent across CIFAR-10, CIFAR-100, and ImageNet: the confidence term receives the highest weight, attribution relevance the next, and instability the smallest (negative) contribution. RQ3 shows that perturbing thresholds by ±10% changes accuracy by <0.3% and exit distributions by <2%, indicating that IEES is not sensitive to precise tuning and generalizes well without repeated calibration.
>
>
> **6. “What if Grad-CAM is replaced by another attribution method?”**
> IEES does not rely on Grad-CAM specifically; it requires only a per-exit attribution map. PFAM is instantiated with Grad-CAM in our experiments for computational efficiency and ease of integration, but attribution strength and stability can be computed using any generator, including Integrated Gradients, LayerCAM, Grad-CAM++, or Occlusion. Evaluating IEES with alternative attribution methods is already noted as future work in the Limitations section.
>
> We thank the reviewer again for the helpful feedback.

---

### Official Review · Reviewer_diB3 · 2025-10-31

**Soundness:** 2
**Presentation:** 3
**Contribution:** 2
**Rating:** 2
**Confidence:** 3

**Summary:**

This submission introduces IEES, an exit policy for early-exit image classification models. The proposed approach first designs a novel metric inspired from the explainable AI domain and based on semantic reliability, that combines commonly used confidence with class-relevant activation strength and stability in attribution focus. At deployment time the proposed metric is approximated through a random forest predictor, to reduce the computational burden of attribution computation.

**Strengths:**

- Revisiting the exit policy in input-dependent computation approaches is a very timely and interesting problem.
- The proposed unification of efficiency and explainability through the proposed attribution-stability based metric (combined with previous confidence solutions) is insightful and demonstrated effectiveness in the experimental evaluation under the examined assumptions.
- The manuscript is very polished and easy to follow.

**Weaknesses:**

1. Some of the design choices of the proposed approach are not adequately justified. For example, since the proposed metric is based on stability, why does the proxy predictor in Eq.7 only consider features from the current exit as inputs?
2. Given the fact that a predictor is trained to serve as exit policy (approximating the proposed metric), it is unclear why the authors chose to go with prediction of a semantic reliability metric, rather than direct prediction of the early-exit decision itself. A clear comparison between the two is required to justify the contribution of this work in the online (inference-time) setting.
3. Since stability is a key property of the proposed exit policy, an obvious baseline would be to consider the stability of logits, or predictions between successive early exits, rather than the proposed attribution-based metric. Further comparisons are required to demonstrate the effectiveness of the proposed components in this context.
4. Although the proposed results showcase the effectiveness of the proposed early exit scheme across some CNN models and image classification datasets, an application of the proposed approach to Transformer architectures for the same tasks is important to showcase the generality of the proposed components and applicability to SoTA approaches.
5. Although briefly demonstrated in the appendix, further experiments are required to compare the proposed metric with its proxy prediction through random forests in terms of out-of-distribution sample handling and other important properties that the original metric would be able to capture, but the proposed approximation is most likely not exposed to during training.

**Questions:**

Please consider replying on the comments raised above.

---

> ### Author Response · Authors · 2025-11-20
>
> We thank the reviewer for the thoughtful and constructive feedback. Below we address all concerns with added clarification.
>
> **1. Why does the proxy use only current-exit features?**
> The proxy (Eq. 8) is designed as a lightweight exit-local approximation of the IEES oracle. IEES (Eq. 5) already encodes attribution-based stability across exits, and this stability signal is included in the target during proxy training. Using only features available at the current exit keeps inference efficient while still allowing the proxy to approximate the full IEES score.
>
> **2. Why regress the IEES score instead of predicting exit/continue?**
> Regressing the continuous IEES value preserves the full reliability structure captured by the metric and enables post-hoc threshold selection for different accuracy–latency operating points. A binary exit/continue classifier would fix a single operating point and discard the semantic-reliability information encoded in IEES.
>
> **3. Why not use logit stability as a baseline?**
> We agree that logit-level consistency is a natural baseline. In response to the suggestion, we added a logit-stability variant in the revision (Appendix I.1). While this baseline offers small gains over tuned confidence thresholds, it consistently underperforms IEES across all datasets. Logit similarity measures only prediction-level agreement, whereas IEES quantifies agreement in the underlying explanatory evidence. The ablation shows that explanation-level stability offers a more reliable routing signal, with IEES improving overall accuracy by approximately +0.5–2.0% over logit-stability.
>
> **4. Applicability to Transformers.**
> We agree with the reviewer. Although our experiments use CNN-based multi-exit architectures for controlled analysis of attribution progression, IEES is model-agnostic: none of its components assume convolutional structure. PFAM can also be instantiated for Transformers using standard attribution tools (e.g., attention rollout, gradient-based methods). As noted in the Limitations section, extending IEES to ViTs and multimodal Transformer architectures is part of our ongoing work.
>
>
> **5. Proxy vs. IEES oracle under distribution shift.**
> Appendix B evaluates the proxy’s fidelity to the IEES oracle across CIFAR-10, CIFAR-100, and ImageNet, showing strong agreement (R² > 0.80, ρ > 0.85, MAE < 0.06). We also evaluate robustness under distribution shift: on CIFAR-10-C, proxy-based exits differ by < 2% from those obtained using the attribution-based IEES oracle, and final accuracy differs by < 0.3%, even though the proxy is trained only on clean data and without attribution features. These results indicate that the proxy preserves the decision structure of the IEES metric—even under shift—while providing an efficient inference-time surrogate. IEES remains the reference metric capturing attribution stability and semantic reliability.
>
> We thank the reviewer again for the valuable feedback.

---

### Official Review · Reviewer_TTQm · 2025-11-01

**Soundness:** 2
**Presentation:** 3
**Contribution:** 2
**Rating:** 2
**Confidence:** 3

**Summary:**

The work presents an explainability-aware early exit framework that built using three components: 1) Progressive feature attribution maps that visualize how model focus evolves across exits, producing localized and cumulative attributions. 2) It also provides an interpretability-based Early-Exit score (IEES) that quantifies the exit reliability by unifying model confidence, feature relevance and attribution stability. 3) Finally the method trains a lightweight IEES predictor that can provide a pseudo-IEES score during inference. The method is backed by experimental analysis on ResNet-18, MSDNet and MobileNetV3 architectures.

**Strengths:**

1) The method proposes an explainability-aware early exit framework, which is important.

2) The IEES scores proposed in the paper seem to be a reliable metric for making an exit decision, which makes the prediction reliable and trustworthy.

3) The paper is well written and easy to follow.

**Weaknesses:**

1) Multiple baselines that work on improving the trustworthiness of EE frameworks are missing [1], [2], [3], [4], [6], they should not only be cited but also be compared.

2) The method heavily depends on the validation dataset, as it sets multiple hyperparameters using that, it restricts the generalization of the method. Also, as the method has a number of exits + (w_1, w_2, w_3) as hyperparameters, the grid search seems extremely costly. Also, the method trains the lightweight IEES score predictor on the validation set, which again tells of the dependence on the validation set.

3) The authors only experiment with the image classification tasks, which is a major concern as the performance needs to be shown on the recent language generation and other GenAI models to make the work worthy.

4) The paper does not provide any model training details what loss function to use, what weights to give for exits, etc., is left for the reader to assume.

5) The author missed multiple relevant papers, some of them are below:


[1] https://papers.neurips.cc/paper_files/paper/2022/file/6fac9e316a4ae75ea244ddcef1982c71-Paper-Conference.pdf

[2] https://proceedings.neurips.cc/paper_files/paper/2024/file/ea5a63f7ddb82e58623693fd1f4933f7-Paper-Conference.pdf

[3] https://openaccess.thecvf.com/content/WACV2024/papers/Meronen_Fixing_Overconfidence_in_Dynamic_Neural_Networks_WACV_2024_paper.pdf

[4] https://aclanthology.org/2024.findings-acl.101.pdf

[5] https://aclanthology.org/2023.emnlp-main.362.pdf

**Questions:**

1) How authors decide the layers in which exits are attached. Also, based on the design, the method chooses different thresholds for each exit and performs a grid search to fix them; it will become infeasible when the model becomes large (32-layer language model). Any thoughts?

2) A suggestion will be, please don't write sentences like "provide little justification for their choices" (line 015), methods usually develop scores which are theoretically grounded.

---

> ### Author Response · Authors · 2025-11-20
>
> We thank the reviewer for the thoughtful and constructive feedback. Below we address all concerns with added clarification.
>
> **1. Missing trustworthy EE baselines.**
> We appreciate these references and agree that extending early-exit methods to language and generative models is an important direction. Table 2 surveys multi-exit classification models that introduce an explicit exit-decision mechanism. Within this scope, CEEBERT aligns well and has been added with its reported accuracy and speedup. The other suggested works relate to areas such as safe decoding, calibration, overconfidence mitigation, and risk-controlled generation. While these are valuable contributions, they address objectives different from the multi-exit classification setting considered in Table 2. For completeness, we now cite and briefly discuss them in the revised Related Work section.
>
> **2. Dependence on validation set / grid search cost.**
> IEES calibrates three global weights and one threshold per exit using a coarse grid search on a held-out validation split. This requires only forward passes and no retraining, making it lightweight and consistent with standard early-exit practice (BranchyNet, MSDNet, DeeBERT, CEEBERT). The proxy predictor is trained on the training set (with a small hold-out only for tuning), not the validation set. Moreover, thresholds calibrated on CIFAR-10 transfer well to CIFAR-10-C (Sec. 5.3), indicating IEES does not overfit the validation split and generalizes reliably under distribution shift.
>
> **3. Only image classification tasks.**
> We focus on CNN-based image classification to analyze attribution progression and exit behavior in a controlled setting. However, IEES is model-agnostic: its components (confidence, attribution relevance, stability) do not rely on convolutional structure and extend naturally to ViTs, BERT-style models, and non-vision domains. Extending IEES/PFAM to ViTs, transformer architectures, and other modalities is already highlighted as an important future direction in the Limitations section, and doing so would further strengthen the work.
>
> **4. Training details not provided.**
> All training details—including optimizer, learning rate, schedule, loss formulation, exit-wise weights, augmentations, and number of epochs—are given in Section 4 (Experimental Setup). All exits and the backbone are trained jointly using the standard weighted exit-wise cross-entropy loss used in BranchyNet, and we have added an explicit pointer from Section 3 to Section 4 to make this easy to locate in the revised manuscript.
>
> **5. Missing relevant papers.**
> Thank you for pointing out these works. We have now incorporated all the suggested papers into the revised manuscript. CEEBERT and DeeBERT have been added to the comparison table, and all other referenced works are cited and discussed in the Related Work section.
>
> **Questions**
> **Q1 — How do authors decide the exit layers?**
> Exit locations follow the placement used in prior multi-exit works such as BranchyNet and MSDNet, where exits are added after major semantic blocks. Our goal in this paper is to evaluate exit policies rather than propose new backbone designs, so we follow these commonly used placements to ensure comparability across methods. IEES itself is compatible with any exit-placement strategy.
>
> **Q1 (continued) — Grid search and deep models (e.g., 32-layer LLM).**
> IEES threshold calibration follows the same protocol used in BranchyNet, MSDNet, and DeeBERT: one threshold per exit tuned on a validation split using a coarse grid. This procedure scales with the number of exits rather than model depth, and therefore remains feasible even for deep architectures. Calibration requires only forward passes and no retraining, keeping it lightweight and practical for large models, including Transformers.
>
> **Q2 — Wording suggestion**
> Thank you for the suggestion. Our intention was not to imply that existing exit policies lack theoretical grounding. Confidence- and entropy-based exit criteria are well studied and widely validated in the early-exit literature. The point we aimed to convey in the abstract concerns interpretability: even when these methods are principled, they typically do not explain why a particular sample exits early. To avoid any unintended implication, we have revised the abstract to:
> “Conventional exit policies—often based on confidence thresholds—make effective decisions but do not inherently provide explanations for why a sample exits early.”
>
> We sincerely thank again  the reviewer for the detailed and thoughtful feedback.

---

### Official Review · Reviewer_X2q6 · 2025-11-01

**Soundness:** 3
**Presentation:** 3
**Contribution:** 3
**Rating:** 6
**Confidence:** 5

**Summary:**

This paper proposes IEES (Interpretability-based Early Exit Score), a method for determining when to terminate inference in early-exit neural networks. Instead of relying only on prediction confidence, IEES integrates three complementary cues into a single decision score: prediction confidence, class-weighted activation strength, and attribution stability. The accompanying Progressive Feature Attribution Maps (PFAM) visualize how the model’s semantic focus evolves across exits. Experiments on CIFAR-10, CIFAR-100, and ImageNet show improved accuracy–latency trade-offs over confidence-based policies and competitive performance with recent early-exit baselines.

**Strengths:**

+ Clear motivation for combining multiple cues (confidence, activation, and stability) to guide exit decisions.
+ Good experimental evidence across three datasets, including ablations and robustness tests under threshold perturbation and distribution shift.
+ Quantitative evaluation of interpretability via faithfulness and alignment metrics, and introduction of PFAM as an intuitive visualization tool.
+ Well-written and well-structured paper with clear definitions and improved presentation quality.

**Weaknesses:**

- The evaluation scope remains limited to CNN-based image classifiers; no results for ViTs, transformers, or non-vision domains.
- The claim of “anytime prediction” is only approximate since exits occur at discrete layers.
- The computational overhead of attribution computation is not quantified, leaving the efficiency benefit somewhat incomplete.
- Comparisons could include more diverse dynamic routing methods (e.g., RANet, DeeBEE) to better contextualize IEES performance.
- Some hyperparameter tuning details (e.g., validation protocol for threshold search) could be explained more precisely.
- A highly relevant work is missing from related work: "Early-exit Convolutional Neural Networks" by Demir et al.
- Fonts in plots (Fig2& Fig3) are too small.
- I am not sure why evaluation on domain-shift needed.

Minor:
- It is not clear what f is in Eq.1.
- L49: I think the definition of F_i(x) is important to understand the method. It should be moved to the main text from Appendix.
- "standard augmentation" should be clarified (L201).
- In Table 1, are the numbers under E1...E4 accuracies of examples exiting that specific gate? This is not clear from the caption of main text.

**Questions:**

- Since there are only three or four hardcoded exits, why do you call it "anytime prediction"?
- L214: What is the point of comparing PFAM with GradCAM? Isn't PFAM Grad CAM as mentioned in Eq.2?
- Table 1 shows that early-exit accuracy is better than no-early-exit accuracy. While authors provide an explanation in Section 5.1, this is not a consistent result in the literature (as also evidenced in Table 2 column \delta accuracy). Can you explain this further or provide more visual results to support your explanation (e.g. showing that examples exiting at later gates are more complicated examples)?

---

> ### Author Response · Authors · 2025-11-20
>
> We thank the reviewer for the constructive feedback. Below we summarize clarifications and revisions made in response.
> **1.Scope focused on CNN classifiers.**
> Our experiments use CNNs to enable controlled analysis of attribution progression and exit stability. IEES, however, is model-agnostic: its components—confidence, attribution relevance, and attribution stability—do not depend on convolutional structure. As noted in the Limitations section, extending IEES/PFAM to ViTs, transformer models, and non-vision domains is an important direction for future work.
>
> **2. Use of the term “anytime prediction”.**
> Our usage is consistent with how the term appears in prior early-exit work (e.g., MSDNet; Zero-Time-Waste), where ‘anytime prediction’ refers to producing predictions at multiple computational budgets even though the exits are discrete.  To avoid ambiguity, we have added the following to §7:
> “We use the term anytime prediction in its standard sense, as in MSDNet and Zero-Time-Waste—denoting the ability to produce valid predictions at multiple computational budgets—even though exits are placed at discrete layers.”
>
> **3. Attribution overhead .**
> Appendix B reports that attribution computation costs ~0.4–0.5 ms/sample, while the proxy requires <0.1 ms/sample and removes attribution entirely at inference time. We have now clarified this in §3.4:
> “For reference, Appendix B also reports that computing attribution maps costs approximately 0.4–0.5 ms per sample, whereas the learned proxy requires less than 0.1 ms per sample and eliminates all attribution overhead during inference.”
>
> **4. Comparisons to RANet, DeeBERT/DeeBEE...**
> Table 2 is scoped specifically to multi-exit classification models that introduce explicit exit-decision mechanisms. Following the reviewer’s suggestion, we added DeeBERT and CEEBERT. RANet is not included because it performs dynamic routing without intermediate classifiers or threshold-based exits.
>
> **5. Hyperparameter tuning detail.**
> The threshold–validation protocol is already described in §3 and §4. For ease of reference, we have added a brief summary of the tuning procedure in Appendix L.
>
> **6. Missing citation (Demir & Akbas, 2024).**
> We added this architectural early-exit CNN paper to §2:
> “ Demir & Akbas (2024) enhance CNN-based early exits with learnable confidence branches and a unified accuracy–cost objective.”
>
> **7. Plot font sizes.**
> We will enlarge the fonts in Figs. 2–3 in the final version.
>
> **8. “Domain shift” phrasing.**
> The experiment evaluates robustness under distribution shift (CIFAR-10 → CIFAR-10-C), not domain shift. We have updated the manuscript accordingly and renamed the subsection to ‘Distribution Shift Evaluation’ in §5.3.
>
> **Minor Clarifications**
>
> **(a) Definition of fi​ in Eq. 1.**
> Clarified in the revised paper as:
> “fi​ denotes the backbone up to exit i, mapping the input x to the intermediate feature representation used by that exit.”
>
> **(b) Definition of Fi​(x).**
> We agree that this term is central. We have moved its full formulation from the Appendix into §3.3.
>
> **(c) “Standard augmentation.”**
> For CIFAR-10/100: random crop with 4-pixel padding + random horizontal flip.
> For ImageNet: Random Resized Crop (224×224) + horizontal flip. This is now stated in §4.
>
> **(d) Table 1 (E1–E4 accuracies).**
> We clarified in the caption that each number is the accuracy for samples exiting specifically at that exit.
>
> **Questions**
>
> **Q1: Why call it “anytime prediction” with 3–4 exits?**
> As noted above, we follow standard usage in MSDNet and Zero-Time-Waste. To avoid confusion, we adopt the phrasing “discrete anytime prediction.”
>
> **Q2: Why compare PFAM with Grad-CAM if PFAM uses Grad-CAM?**
> PFAM is not an attribution method; it is a progressive multi-exit construction that sits on top of a base generator (Grad-CAM in our experiments). In Fig. 3, the baselines (Grad-CAM, IG, SmoothGrad, Occlusion) show independent per-exit maps computed in isolation. PFAM instead produces progressive, depth-aligned attribution trajectories, reusing shared activations and enforcing consistent normalization across exits. The purpose of the comparison is to illustrate how PFAM improves the structure and clarity of any underlying attribution method, not to compare PFAM against its own generator.
>
> **Q3. Early-exit accuracy > full-depth accuracy...**
> Early exits do not always outperform full-depth models; this is why we state the effect occurs “in some cases.” The small gains we observe arise from (1) joint multi-exit training, which provides mild deep supervision and can improve generalization, and (2) IEES routing, which avoids premature shallow exits by sending semantically unstable samples to deeper ones. Figures 1, 3, and Appendix 9–11 show that early-exit samples have stable, focused attributions, whereas visually complex or representation-level difficult inputs exhibit unstable early maps that stabilize only at deeper exits. Thus, later exits naturally handle these harder cases

---

### Note · Program_Chairs · 2026-01-17
**Submission Desk Rejected by Program Chairs**

The following references in this submission do not refer to real documents and/or have major errors in bibliographic information:

 Risk-controlled early exiting for deep neural networks. In Advances in Neural Information 2024.